# Theory of Quantum Path Entanglement and Interference with Multiplane Diffraction of Classical Light Sources

**DOI:** 10.3390/e22020246

**Published:** 2020-02-21

**Authors:** Burhan Gulbahar

**Affiliations:** Department of Electrical and Electronics Engineering, Ozyegin University, Istanbul 34794, Turkey; burhan.gulbahar@ozyegin.edu.tr

**Keywords:** multiplane diffraction, entangled histories, quantum path entanglement, quantum path interference, Leggett–Garg inequality

## Abstract

Quantum history states were recently formulated by extending the consistent histories approach of Griffiths to the entangled superposition of evolution paths and were then experimented with Greenberger–Horne–Zeilinger states. Tensor product structure of history-dependent correlations was also recently exploited as a quantum computing resource in simple linear optical setups performing multiplane diffraction (MPD) of fermionic and bosonic particles with remarkable promises. This significantly motivates the definition of quantum histories of MPD as entanglement resources with the inherent capability of generating an exponentially increasing number of Feynman paths through diffraction planes in a scalable manner and experimental low complexity combining the utilization of coherent light sources and photon-counting detection. In this article, quantum temporal correlation and interference among MPD paths are denoted with quantum path entanglement (QPE) and interference (QPI), respectively, as novel quantum resources. Operator theory modeling of QPE and counterintuitive properties of QPI are presented by combining history-based formulations with Feynman’s path integral approach. Leggett–Garg inequality as temporal analog of Bell’s inequality is violated for MPD with all signaling constraints in the ambiguous form recently formulated by Emary. The proposed theory for MPD-based histories is highly promising for exploiting QPE and QPI as important resources for quantum computation and communications in future architectures.

## 1. Introduction

Quantum temporal correlations are analyzed with diverse methods by utilizing histories or trajectories of evolving quantum systems with more recent emphasis on mathematical formulation of the entangled superposition of quantum histories in Reference [1], i.e., denoted with the entangled histories framework. These varying methods include Feynman’s path integral (FPI) formalism [2] as the most fundamental of all inherently including histories, consistent histories approach defined by Griffiths [3,4,5], and the recently formulated entangled histories framework [1] and two-state vector formalism [6,7] while all emphasizing correlations in time as standard quantum mechanical (QM) formalisms without violating Copenhagen interpretations. Multiplane diffraction (MPD) design as a simple linear optical system was recently proposed for quantum computing (QC) [8,9] and for modulator design in classical optical communications [10] by exploiting the tensor product structure of quantum temporal correlations as quantum resources while utilizing only the classical light sources and conventional photon-counting intensity detection. The MPD architecture generates interference of an exponentially increasing number of propagation trajectories along the diffraction events through multiple slits on the consecutive planes. The simplicity of source and detection in MPD setup combined with the highly important promise of the utilization of the tensor product structure of the temporal correlations as quantum resources motivates the definition and study of quantum trajectories or histories in MPD as novel quantum resources. These new resources denoted as *quantum path entanglement* (QPE) and *quantum path interference* (QPI) are defined and theoretically modeled in this article in terms of the temporal correlations and interference among the trajectories, respectively, to be exploited for future quantum computing and communications systems.

In this article, MPD design is, for the first time, proposed for defining QPE and QPI as novel quantum resources. Operator theory modeling for MPD-based resources is presented by combining the consistent histories approach of Griffiths [1,3,4,5] and the entangled histories framework in Reference [1] with the FPI approach as the inherent structure of MPD creating Feynman paths. MPD creates quantum propagation paths through individual slits in a superposition in which the linear combinations result in evolving quantum history states. It has low experimental complexity with classical light sources and conventional photon-counting detection for near-future experimental verification. The theory of QPE and QPI based on MPD proposed in this article provides a set of tools to explore new structures composed of the correlations and interference among the paths for future applications in quantum computing and communications and provides QM foundational studies based on quantum histories.

The concept of the entangled histories is defined in References [1,11] as the quantum history which cannot be described as a definite sequence of states in time. There is a superposition of multiple timelines of sequences of events. In this article, we follow similar terminology and denote the temporal correlation among the quantum propagation paths unique to the MPD design with QPE, i.e., emphasizing the entanglement among the path histories similar to References [1,11]. Tensor product structure among the temporal correlations of multiple time instants is utilized as a novel resource for computing in References [8,9] and for communications in Reference [10] in an analogical manner to the multiparticle spatial correlations of the conventional quantum entanglement resources. MPD provides a simple system design inherently including such states having correlations among the paths denoted with QPE. A concrete example of a history state in MPD composed of diffraction events through *N* planes is defined as follows:(1)$$\sum_{n}\pi_{n}\left[P_{N,s_{n,N}}\right]⊙\left[P_{N-1,s_{n,N-1}}\right]⊙\ldots ⊙\left[P_{1,s_{n,1}}\right]⊙\left[\rho_{0}\right]$$
where $P_{j,s_{n,j}}$ is the projection operator for diffraction through the slit indexed with $s_{n,j}$ on *j*th plane and for *n*th trajectory, $\pi_{n}$ as 0 or 1 allows to choose a compound set of trajectories, ⊙ denotes tensor product operation, and $\left[\rho_{0}\right]$ denotes the initial state. The quantum state of the light after diffraction through consecutive *N* planes includes a superposition of different trajectories through the slits. Experiments for entangled histories has just been, for the first time, performed in Reference [11] by using the polarization states of a single photon and by creating Greenberger–Horne–Zeilinger (GHZ)-type states. MPD-based design compared with complex single photon setup allows the classicality of light sources and simple intensity detection (or photon counting) as a significantly low complexity tool to study quantum histories and QM foundations with near-future experiments. MPD utilizes simple and widely available coherent sources such as Gaussian wave packets of standard laser output conventionally denoted as classical light.

In this article, an important property of MPD-based QPE is, for the first time, presented: Leggett–Garg Inequality (LGI) violations as the temporal analog of Bell’s inequality. One of the fundamental tools to analyze quantum temporal correlations of a system is to check the violations of LGIs [12]. LGIs, as proposed by Leggett and Garg in 1985, check a system in terms of the fundamental principles of macroscopic realism (MR) and noninvasive measurability (NIM) such that the systems obeying these rules satisfy the intuition about the classical macroscopic world [13]. QM systems violate LGIs such that MR principles implying the existence of a preexisting value of a macroscopic system and the NIM principle implying the measurement of the value without disturbing the system are both invalidated [14,15]. LGI violations [12,13,14,16,17] are utilized for various purposes such as testing temporal correlations of a single system as an indicator of the quantumness and analyzing QC systems, e.g., Grover’s algorithm violating temporal Bell inequality [18]. The simple LGI inequality with three-time formulation violated with various QM setups is defined as follows:(2)$$C_{01}+C_{12}-C_{02}\leq 1$$
where $C_{ij}\equiv \langle Q_{i}Q_{j}\rangle$ is the expected value of the multiplication of the dichotomic observables $Q_{i}$ as the measurement outcomes at time $t_{i}$. The left-hand side is maximally violated by QM systems with the value of 3/2. The violation analysis of LGIs is, for the first time, performed for MPD by utilizing the recently proposed ambiguous form by Emary in Reference [16] with the precautions regarding the signaling-in-time (SIT) problem in order to convince a macrorealist about the noninvasive nature of measurements, i.e., to prevent signaling forward in time with measurements. This is achieved by inferring event probabilities from ambiguous measurements rather than direct measurements and by modifying the fundamental inequality in Equation (Equation 2) by including a signaling term and by providing a NIM-free bound as described in detail in the Results section. The violation of LGI with no-signaling assumption reaching >0.2, i.e., left-hand side of >1.2, is numerically obtained for three-time formulation of LGI in MPD setup. The optimization study to maximize it to the calculated bounds [16] is left as an open issue. Besides that, a novel system design, i.e., MPD, violating LGIs with classical light sources is proposed in this article, complementing the recent experimental result in Reference [19] utilizing linear polarization degree of freedom of the classical light to violate LGIs. However, MPD utilizes photon-counting intensity detection with a significantly low experimental complexity. It is also simpler compared with the LGI violating architectures utilizing single-photon sources and Mach–Zehnder interferometers [11,20,21]. Besides that, light sources not fully coherent in terms of spatial and temporal dimensions are theoretically modeled while the violation of LGI and QPI are numerically analyzed for specific MPD setup geometry satisfying coherence of light under Gaussian source beam assumptions.

On the other hand, LGIs are interpreted in a quantum contextual framework in Reference [22], where the contextuality implies the impossibility to consider a quantum measurement as revealing a preexisting property independent of the set of measurements. It is also analyzed in relation with consistent histories approach in Reference [23]. Furthermore, nonlocality and contextuality are presented as important quantum resources [24]. Therefore, the relation of the proposed QPE and QPI resources with quantum contextuality is an open issue to be explored.

The other important property of quantum histories is the interference among them denoted by QPI. To the best of the author’s knowledge, theoretical modeling of the interference among quantum history states leading to a counterintuitive observation to be easily verified experimentally has not been previously formulated. Implementation of the theoretically modeled QPI setup will significantly improve our understanding about QM fundamentals regarding time. QPI is the temporal analogue of the spatial interference obtained in Young’s double-slit setup. Destructive and constructive interferences among the paths are observed in the time domain for the QPI case. A special case is modeled such that decreasing the number of photons to diffract through a plane by removing a Feynman path results in an increase in the number of photons diffracting through the next plane due to the interference between two quantum trajectories. This is proposed, for the first time, as a counterintuitive nature of the interference among the quantum histories.

The novel contributions of the article are summarized as follows:introduction and operator theory modeling of two novel quantum resources, i.e., QPE and QPI, denoting temporal correlations and the interference among quantum trajectories, respectively, in MPD while utilizing the tensor product structure for future quantum computing and communication architectures and foundational QM studies;operator theory modeling of MPD-based resources QPE and QPI by combining history-based previous formulations of quantum histories [1,3,4,5] with FPI formalism;theoretical modeling and numerical analysis of MPD setup for the violation of LGI, with the ambiguous and no-signaling forms recently proposed by Emary in Reference [16], reaching >1.2 of correlation amplitude numerically obtained for three-time formulation while leaving the maximization of the violation to the boundary levels as an open issue;a novel setup, i.e., MPD, violating the ambiguous form of LGI with classical light sources complementing the recent experiment utilizing linear polarization degree of freedom of the classical light [19] while MPD setup with remarkably low complexity design utilizing classical light sources and photon-counting intensity detection;theoretical modeling and numerical analysis of counterintuitive properties and examples of the interference among MPD-based Feynman paths denoted as QPI promising to be easily verified experimentally in future studies;the modeling and numerical analysis of the coherence properties of the light sources in terms of spatial and temporal dimensions while discussing design issues for MPD setup with coherent light sources; anddiscussion for future applications of QPE and QPI as quantum resources and experimental implementations.

The paper is organized as follows. We firstly define MPD setup with diffractive projection and measurement operators in Section 2.1 and Section 2.2. It is followed by the history state modeling of QPE in Section 2.3. Then, we present theoretical modeling of the violation of LGI in Section 2.4, followed by QPI scenario in Section 2.5. Then, numerical analysis is presented in Section 2.6. We provide the conclusions and discuss future applications of QPE and QPI based on MPD setup in Section 3. Finally, the methods utilized for theoretical modeling are presented in Section 4.

## 2. Results

### 2.1. MPD Setup for Quantum Temporal Correlations

MPD setup is formed from N−1 diffraction planes of multiple slits in front of a classical light source and the measurement of interference pattern with *N* sensor planes, i.e., both diffraction and sensing on the same plane, as shown in Figure 1a. It is also possible to locally count the diffracted photons with the measurement planes inserted between the diffraction planes as discussed in Section 2.6. The utilized light source is assumed to be coherent as the closest analog of a classical light field emphasizing the absence of nonclassical states of light such as single photon generation, squeezed light, or multiple particles of entangled photons [25]. The standard laser output is almost perfectly a coherent state corresponding to the fundamental transverse modes of light field distribution producing Gaussian beams. This coherent Gaussian wave function keeps the position and momentum uncertainties stationary as emphasized by Glauber [26]. It is an eigenstate of the annihilation operator $\hat{a}$ for the harmonic oscillator, i.e., $\hat{a}\;|\alpha \rangle =\alpha \;|\alpha \rangle$, represented as follows in the complete orthonormal basis of the number states |n〉 of the single mode oscillator [26]:(3)$$|\alpha \rangle =e^{-{\left|\alpha \right|}^{2}\;/\;2}\sum_{n}\frac{\alpha^{n}}{{\left(n!\right)}^{1/2}}|n\rangle$$
where its representation in the position basis gives the Gaussian form. Therefore, the source is assumed to have normalized Gaussian wave function $\Psi_{0}\left(x_{0}\right)\equiv \exp\left(-x_{0}^{2}/\left(2\sigma_{0}^{2}\right)\right)/\sqrt{\sigma_{0}\;/\;\sqrt{\pi }}$ with the standard deviation term $\sigma_{0}$.

Each plane is assumed to be capable of performing measurement with photodetectors for counting the number of photons hitting the detector area. Therefore, a plane either allows projective diffraction of light through slits denoted by the operator symbol P or performs measurement denoted by M on its sensor array positions where there are no slits. Gaussian slits are utilized with FPI modeling for simplicity [2,8] as mathematically described in Equation (Equation 7) in the next subsection. Light is assumed to perform free space propagation between consecutive planes. The plane with the index *j* has $S_{j}$ slits, where the central positions and widths of slits are denoted by $X_{j,i}$ and $W_{j,i}$, respectively, and j∈[1,N−1] and $i\;\in [1,S_{j}]$. The widths of the slits are assumed to be the same on each plane but not constrained among different planes. Distance between the *i*th and *j*th planes is denoted by $L_{i,j}$, where the distance from the light transmitter source to the first plane is given by $L_{0,1}$. Light is assumed to have propagation in the *z*-axis with the velocity given by *c*, while quantum superposition interference is observed in the *x*-axis as a one-dimensional model which can be easily extended to two dimensions (2D) [8]. Interplane distances and durations are denoted by the vectors ${\vec{L}}^{T}=\left[L_{0,1}\;\ldots \;L_{N-1,N}\right]$ and ${\vec{t}}^{T}=\left[t_{0,1}\;\ldots \;t_{N-1,N}\right]\equiv {\vec{L}}^{T}\;/\;c$, respectively, where transpose is denoted by ${\left(.\right)}^{T}$. The value ${\vec{t}}^{T}$ is accurate with the assumption $L_{j-1,j}\gg W_{j,i},X_{j,i}$ for j∈[1,N−1] and $i\in [1,S_{j}]$ such that QM effects are emphasized in the *x*-axis. Nonrelativistic modeling is assumed. We do not consider the effects of environment dephasing or decohering of the interference pattern for double-slit setups [27,28]. Furthermore, minor effects of exotic paths [29] on the numerical results are ignored as discussed in Reference [8] without affecting the main modeling.

Free-particle evolution kernel for the optical propagation paths between time–position values $\left(t_{j},x_{j}\right)$ and $\left(t_{j+1},x_{j+1}\right)$ is defined as follows [9,10] with the same form for electron propagation [2,8]:(4)$$K\left(x_{j+1},t_{j+1};x_{j},t_{j}\right)=\sqrt{\frac{m}{2\;\pi \;ı\;\hbar \;\Delta t}}\;\exp\left(\frac{ı\;m\;\Delta x^{2}}{2\;\hbar \;\Delta t}\right)$$
where $\Delta t=t_{j+1}-t_{j}$, $\Delta x=x_{j+1}-x_{j}$, m≡ℏk/c is the virtual mass term for the photon with the wave number k=2π/λ, and λ is the wavelength of the light.

The validity of Fresnel diffraction formulation for quantum optical propagation is verified based on recent experimental [30] and theoretical [31] studies, while Fourier optics [32] extension of MPD is recently proposed in Reference [9]. Therefore, the Fresnel diffraction integral for free space proposed in Equation (Equation 4) and its consecutive application with FPI formalism are theoretically valid and highly reliable for the simple design of MPD. The proposed theoretical model significantly promises to be verified with near-future experiments due to the simplicity of the setup. Then, the propagated wave function $|\Psi_{j}\rangle =\int_{-\infty }^{\infty }dx_{j}|x_{j}\rangle \Psi_{j}\left(x_{j}\right)$ on the *j*th plane becomes as follows by utilizing Equation (Equation 4) consecutively in FPIs [8,9,10]:(5)$$\Psi_{j}\left(x_{j}\right)\;\equiv \;\sum_{n=0}^{N_{j}-1}\psi_{j,n}\left(x_{j}\right)\;\equiv \;\sum_{n=0}^{N_{j}-1}\left(Y_{j}\;e^{\left(A_{j-1}\;+\;ı\;B_{j-1}\right)\;x_{j}^{2}}\;e^{{\vec{x}}_{n}^{T}\;H_{j-1}\;{\vec{x}}_{n}}\;e^{\left({\vec{c}}_{j-1}^{T}\;+\;ı\;{\vec{d}}_{j-1}^{T}\right){\vec{x}}_{n}\;x_{j}}\right)$$
where $\psi_{j,n}\left(x_{j}\right)$ is the contribution for each *n*th propagation path through the slits on the overall superposition, and the definitions of the notations *n* and $N_{j}$ are explained next while $Y_{j}=\chi_{0}\;\left(\prod_{l=1}^{j-1}\sqrt{\xi_{l}}\right)$; the constants $A_{j-1}$, $B_{j-1}$, $\chi_{0}$, and $\xi_{l}$ for l∈[1,j−1]; $H_{j-1}=H_{R,j-1}\;+\;ı\;H_{I,j-1}$; and the vectors ${\vec{c}}_{j-1}$ and ${\vec{d}}_{j-1}$ depending on the group of {ℏ, *m*, $\sigma_{0}$, $t_{l,l+1}$, and $\beta_{l}\}$ for l≤j−1 are explicitly defined in Reference [8]. Explicit forms of the parameters required for double- and triple-plane setups are provided in Section 4 while formulating LGIs and QPI, respectively, in the following discussions. The total number of paths just before diffraction on the *j*th plane is calculated by $N_{j}=\prod_{l=1}^{j-1}S_{l}$, while the set of slit positions for the path indexed with $n\in [0,N_{j}-1]$ is denoted by ${\vec{x}}_{n}\equiv {\left[X_{1,s_{n,1}}\;\;X_{2,s_{n,2}}\;\;\ldots \;\;X_{j-1,s_{n,j-1}}\right]}^{T}$ while each *n*th path is indexed by the set of diffracted slits as the following:(6)$$Path_{n}\equiv \left\{s_{n,1},\;s_{n,2},\;\ldots \;s_{n,j-1};\;s_{n,l}\in \left[1,S_{l}\right],l\in [1,j-1]\right\}$$
where the specific slit on *l*th plane for *n*th trajectory is indexed with $s_{n,l}$. The same symbol of the position vector $\vec{x}$ is used for both the dimensions *N* and *j*. The size of the vector is inferred from the index of the current plane analyzed throughout the text. The position on the *j*th plane is denoted by $x_{j}$. In Equation (Equation 5) for j=N, each path reaching the *N*th plane is indexed by *n* for $n\in [0,N_{p}-1]$ as shown in Figure 1b for a simple example of $N_{p}=4$, where total number of paths is given by multiplying the number of slits on each plane as $N_{p}\equiv \prod_{j=1}^{N-1}S_{j}$. The vector ${\vec{x}}_{n}\equiv$$[X_{1,s_{n,1}}\;X_{2,s_{n,2}}$…$X_{N-1,s_{n,N-1}}{]}^{T}$ denotes the set of slit positions ordered with respect to the plane indices for *n*th path for the case of *N* planes. Next, diffraction and measurement operators are theoretically defined by emphasizing the operator algebra of multiplane evolution.

### 2.2. Diffractive Projection and Measurement Operators

Projection operator denotes the light to be in the Gaussian slit in a coarse-grained sense [8,33] as follows:(7)$$P_{j,i}\equiv \int_{-\infty }^{\infty }dx_{j}\;\exp\left(-\frac{{\left(x_{j}-X_{j,i}\right)}^{2}}{2\;\beta_{j,i}^{2}}\right)|x_{j}\rangle \langle x_{j}|$$
where $g_{j,i}\left(x_{j}\right)\equiv \exp\left(-{\left(x_{j}-X_{j,i}\right)}^{2}\;/\;\left(2\;\beta_{j,i}^{2}\right)\right)$ is the slit projection function and the effective slit width is $W_{j,i}\equiv 2\;\sqrt{2}\;\beta_{j,i}$, i.e., leading to a $1\;/\;e^{2}$ drop in the intensity, where j∈[1,N−1] and $i\;\in [1,S_{j}]$. Projectors are mutually exclusive with high accuracy such that slit distances are chosen large enough to satisfy $\exp\left(-{\left(X_{j,i_{1}}-X_{j,i_{2}}\right)}^{2}\;/\;\left(2\;\beta_{j,i_{1}}^{2}\right)\right)\ll 1$ for $i_{1}\neq i_{2}$. Total diffraction through all slits of the *j*th plane has the operator $P_{j}\equiv \sum_{i=1}^{S_{j}}P_{j,i}$. Measurement operators are redefined due to the proposed Gaussian slit design such that trace preserving equality is satisfied, i.e., $M_{j}^{†}\;M_{j}+P_{j}^{†}\;P_{j}=I$, where I is the identity operator and ${\left(.\right)}^{†}$ or ${\left(.\right)}^{H}$ denotes Hermitian or conjugate transpose operation. It is assumed that wave function at time $t=t_{0}$ evolves to $|\Psi_{j}\rangle$ and $|\Psi_{j}^{+}\rangle$ for just before and just after diffraction on the *j*th plane at $t_{j}^{-}$ and $t_{j}^{+}$, respectively. The state of the light at $t_{j}^{+}$ has experienced either $M_{j}$ or $P_{j}$. The measurement operator on the *j*th plane is defined as the following:(8)$$M_{j}^{†}\;M_{j}\equiv I-\left(\sum_{i=1}^{S_{j}}P_{j,i}^{†}\right)\left(\sum_{i=1}^{S_{j}}P_{j,i}\right)\equiv \int_{-\infty }^{\infty }dx_{j}\;\left(1-{\left(\sum_{i=1}^{S_{j}}e^{-\frac{{\left(x_{j}-X_{j,i}\right)}^{2}}{2\;\beta_{j,i}^{2}}}\right)}^{2}\right)|x_{j}\rangle \langle x_{j}|$$
Therefore, if we define the measurement operator in FPI formalism as multiplication of the wave function with $m_{j}\left(x_{j}\right)$ reducing the probability to measure the light while approaching the slit center, then the following is obtained by using Equation (Equation 8):(9)$$|m_{j}\left(x_{j}\right){|}^{2}=1-{\left(\sum_{i=1}^{S_{j}}e^{-\frac{{\left(x_{j}-X_{j,i}\right)}^{2}}{2\;\beta_{j,i}^{2}}}\right)}^{2}$$

There are two different types of detection mechanisms in MPD design denoted by $Rec_{1}$ and $Rec_{2}$. In $Rec_{1}$, all of the planes for j∈[1,N] have detectors measuring the incident light and $Rec_{1}$ is the model proposed in this article forming a complete set of diffractive projection $P_{j}$ and measurement $M_{\tilde{j}}$ operators until the final detector plane *N* for j∈[1,N−1] and $\tilde{j}\in \left[1,N\right]$. In this article, $Rec_{1}$ modeling is utilized to model history-based time evolution of the light. An example is shown in Figure 1b, where there is a total of seven different sets of consecutive events forming a complete set of histories. The proposed setup is modeled compatible with the consistent histories approach defined in Reference [3] or the entangled histories framework in Reference [1]. On the other hand, the receiver type with the sensors only on the final plane is denoted by $Rec_{2}$. In $Rec_{2}$, i.e., the modeling utilized in Reference [8] for QC, only the final intensity distribution or interference pattern on the detector plane is measured. There is either no detection at the time $t_{N}^{+}$ or the light is detected on the final detector plane with the index *N*. An operator denoting no detection is defined as $M_{o}$ to form a complete set for $Rec_{2}$; then, $M_{N}^{†}M_{N}+M_{o}^{†}M_{o}=I$. Next, consistent histories approach is applied for MPD setup.

### 2.3. History State Modeling of QPE

Following the definition of consistent histories [3,4,5] and entangled histories [1], a history state is defined for MPD based on the set of projections $M_{j}$ and $P_{j,i}$ on each *j*th plane for j∈[1,N] and $i\;\in [1,S_{j}]$. History Hilbert space is defined as follows:(10)$$H\equiv H_{N}⊙H_{N-1}⊙\ldots ⊙H_{1}⊙H_{0}$$
where $H_{j}$ denotes the set of projections on planes and ⊙ denotes tensor product operation. Hilbert space until $t_{j}^{+}$ includes both projections $P_{j}$ and $M_{l}$ on the planes with the indices l≤j since the light is detected at some plane until $t_{j}$ or still diffracting through the *j*th plane. A general history state with QPE composed of superposition of trajectories is denoted as follows based on the notation (similar to bra-ket but with different notations of (.| and |.) for the histories corresponding to 〈.| and |.〉, respectively) in Reference [1]:(11)$$|\Psi_{N})=\sum_{n}\pi_{n}\left[O_{n}\left(t_{N}\right)\right]⊙\left[O_{n}\left(t_{N-1}\right)\right]⊙\ldots ⊙\left[O_{n}\left(t_{0}\right)\right]$$
where $|\Psi_{j})$ is some history state between times $t_{0}$ and $t_{j}$ for $t_{j}>t_{0}$, the projector $\left[O_{n}\left(t_{j}\right)\right]$ denotes either of $M_{l}$ or $P_{l,i}$ for l≤j and $i\;\in [1,S_{l}]$, and $\pi_{n}$ as 0 or 1 is some permutation choosing a compound set of histories indexed by *n*. Observe that $t_{j}$ includes measurements $M_{l}$ for l≤j as possible events such that the state does not change after measurement. It also includes events with zero probability such as *j*th plane projection at times not equal to $t_{j}$. Some examples for N=4 are as follows:(12)$$\begin{array}{c} \begin{array}{cc} |\Psi_{4}^{a})\equiv & \left[M_{1}\right]⊙\left[M_{1}\right]⊙\left[M_{1}\right]⊙\left[M_{1}\right]⊙\left[\rho_{0}\right]\\ |\Psi_{4}^{b})\equiv & \left[M_{4}\right]⊙\left[P_{3,2}\right]⊙\left[P_{2,4}\right]⊙\left[P_{1,1}\right]⊙\left[\rho_{0}\right]\\ |\Psi_{4}^{c})\equiv & \left[M_{4}\right]⊙\left[M_{2}\right]⊙\left[M_{2}\right]⊙\left[M_{1}\right]⊙\left[\rho_{0}\right] \end{array} \end{array}$$
The state $|\Psi_{4}^{a})$ shows that the light is detected on the first plane at $t_{1}$ while not changing at consecutive time states, i.e., without diffracting even from the first plane. In $|\Psi_{4}^{b})$, the light is diffracted from the first slit of the first plane at $t_{1}$, then is diffracted from the fourth slit of the second plane at $t_{2}$ and from the second slit of the third plane at $t_{3}$, and is finally measured on the fourth plane. The third example $|\Psi_{4}^{c})$ is a state with zero probability due to the orthogonality of the operators on different planes. A simple example for three planes with two slits is shown in Figure 1b with seven different history states while $N_{p}=4$ of them reach the final detector plane as consecutively diffracted trajectories. History Hilbert space summing to the identity denoted by ${\bar{I}}_{H}$ as the family based upon an initial state and neglecting the histories with zero probability is described as follows [3]:(13)$$\begin{array}{c} \begin{array}{cc} {\bar{I}}_{H}= & \sum_{j=1}^{N}\sum_{i_{j-1}=1}^{S_{j-1}}\sum_{i_{j-2}=1}^{S_{j-2}}\ldots \sum_{i_{1}=1}^{S_{1}}\left({\left[M_{j}\right]}^{⊙\;\alpha }⊙\left[P_{j-1,i_{j-1}}\right]⊙\ldots ⊙\left[P_{1,i_{1}}\right]⊙\left[\rho_{0}\right]\right) \end{array} \end{array}$$
where ${\left[M_{j}\right]}^{⊙\;\alpha }$ denotes α≡N+1−j consecutive measurements of $\left[M_{j}\right]$ on the same plane. This includes all the possible history states and evolution for the light until $t=t_{N}$ starting from $t_{0}$. A chain operator is presented in Reference [1] to define the inner product between history states which maps a history state to an operator. The chain operator provides history states with positive semi-definite inner products. This operator is inherently defined in the MPD system as the free-particle evolution kernel $K(x_{1},t_{1};x_{0},t_{0})$. Assume that the free-particle evolution operator with the notation $U_{j+1,j}$ acts as the bridging operator connecting projections at times $t_{j}$ and $t_{j+1}$. Then, chain operator denoted by $\chi_{t_{j+1},t_{j}}$ for the time duration ($t_{j}$, $t_{j+1}$) is defined as follows: (14)$$\begin{array}{ccc} \chi_{t_{j+1},t_{j}}\left\{\left[P_{j+1}\right]⊙\left[P_{j}\right]\right\} & \overset{1}{=} & P_{j+1}\;U_{j+1,j}\;P_{j}\; \end{array}$$(15)$$\begin{array}{ccc} \chi_{t_{j+1},t_{j}}\left\{\left[M_{j+1}\right]⊙\left[P_{j}\right]\right\} & \overset{2}{=} & M_{j+1}\;U_{j+1,j}\;P_{j}\; \end{array}$$(16)$$\begin{array}{ccc} \chi_{t_{j+1},t_{j}}\left\{\left[M_{j}\right]⊙\left[M_{j}\right]\right\} & \overset{3}{=} & M_{j}\;I\;M_{j}\; \end{array}$$(17)$$\begin{array}{ccc} \chi_{t_{j+1},t_{j}}\left\{\left[O_{n}\left(t_{j+1}\right)\right]⊙\left[O_{n}\left(t_{j}\right)\right]\right\} & \overset{4}{=} & O_{n}\left(t_{j+1}\right)\;I\;O_{n}\left(t_{j}\right)\; \end{array}$$
where $\left[O_{n}\left(t_{j+1}\right)\right]$ and $\left[O_{n}\left(t_{j}\right)\right]$ in $\overset{4}{=}$ denote the cases which are not presented in the first three definitions. I is the identity operator equalizing the consecutive measurements on the same plane, i.e., $M_{j}^{l}=M_{j}$, for any integer *l*. Furthermore, it bridges dynamically not possible history states which have zero probability to occur as discussed in Reference [3]. These include consecutive measurements on different planes such as $\left[M_{j+1}\right]⊙\left[M_{j}\right]$, future projection or measurements at a previous time such as $\left[M_{j}\right]⊙\left[P_{j+1}\right]$, or consecutive sets of the same projector $P_{j}$ at future times such as $\left[P_{j}\right]⊙\left[P_{j}\right]$, where free-space propagation in the *z*-axis prevents this. Then, the compound history state mapped or affected by the chain operator is defined as follows:(18)$$\begin{array}{c} \begin{array}{c} \chi_{t_{N},t_{0}}\left|\Psi_{N}\right)\equiv \sum_{n}\pi_{n}\;O_{n}\left(t_{N}\right)\;V_{N,N-1}\;O_{n}\left(t_{N-1}\right)\;\ldots \;V_{1,0}\;O_{n}\left(t_{0}\right) \end{array} \end{array}$$
where $V_{j+1,j}$ denotes either $U_{j+1,j}$ or I.

Besides that, MPD allows to model and explore varying kinds of superposition of history states and QPEs similar to the specific entangled states discussed in Reference [1] resembling the temporal counterpart of Bell states. For example, entangled history states of the GHZ type is experimentally tested in Reference [11]. It is an open issue to utilize MPD to generate and test such states with important implications and applications based on QPE. Next, probability amplitudes of histories are modeled.

#### Event Probabilities

The probabilities characterize the statistical properties of the measurement of classical light. It is assumed that the probability is proportional to the square of the wave function with Born’s postulate. It is calculated by integrating the number of photons on the detector area at a specific position for a time interval *T* enough to obtain the statistical properties [30,31]. The normalized probability is easy to calculate by measuring the number of photons for each event by forming a histogram and then by dividing the number of photons for the specific event to the total number of source photon counts. The number of photons at a particular position *x* is frequently denoted with the integral element ${\left|\Psi \left(x\right)\right|}^{2}dx$, while ${\left|\Psi \left(x\right)\right|}^{2}$ is denoted as the intensity of the light at the particular position. The probability for the particular history state is found with the positive semi-definite inner product defined as follows: (19)$$\begin{array}{c} \begin{array}{cc} (\Phi_{N}|\Psi_{N})\equiv \; & \mathrm{tr}\{{\left(\chi_{t_{N},t_{0}}\left|\Phi_{N}\right)\right)}^{H}\left(\chi_{t_{N},t_{0}}\left|\Psi_{N}\right)\right)\} \end{array} \end{array}$$
where tr{.} is the trace operation. Assume that two specific elementary history states corresponding to specific diffraction paths indexed with $\tilde{n}$ and $\hat{n}\in \left[0,N_{p}-1\right]$ composing the superposition wave function in Equation (Equation 5) are denoted by $|\psi_{N,\tilde{n}})$ and $|\phi_{N,\hat{n}})$, respectively. These paths include only the diffraction projections at the planes with the indices j∈[1,N−1] denoted by $P_{j,s_{\tilde{n},j}}$ and $P_{j,s_{\hat{n},j}}$, respectively. If the initial state $\rho_{0}=|\Psi_{0}\rangle \langle \Psi_{0}|$ and $M_{N}=I$ are included, then the weight of an elementary diffraction history denoted by the inner product $W_{\tilde{n}}\equiv \left(\psi_{N,\tilde{n}}|\psi_{N,\tilde{n}}\right)$ in Reference [3] becomes the following: (20)$$\begin{array}{ccc} W_{\tilde{n}} & = & \mathrm{tr}\left\{U_{N,N-1}\;P_{N-1,s_{\tilde{n},N-1}}\;\ldots \;P_{1,s_{\tilde{n},1}}\;U_{1,0}\;\rho_{0}\;U_{1,0}^{†}\;P_{1,s_{\tilde{n},1}}^{†}\;\ldots \;P_{N-1,s_{\tilde{n},N-1}}^{†}\;U_{N,N-1}^{†}\;M_{N}\right\} \end{array}$$(21)$$\begin{array}{ccc} & = & \mathrm{tr}\left\{\rho_{0}\;U_{1,0}^{†}\;P_{1,s_{\tilde{n},1}}^{†}\;\ldots \;P_{N-1,s_{\tilde{n},N-1}}^{†}\;U_{N,N-1}^{†}\;U_{N,N-1}\;P_{N-1,s_{\tilde{n},N-1}}\;\ldots \;P_{1,s_{\tilde{n},1}}\;U_{1,0}\;\rho_{0}\right\} \end{array}$$(22)$$\begin{array}{ccc} & = & \int_{x_{N}=-\infty }^{\infty }dx_{N}\;{\left|\psi_{N,\tilde{n}}\left(x_{N}\right)\right|}^{2} \end{array}$$
where the trace is realized with respect to the position, $\mathrm{tr}\{\rho_{0}^{2}\}=1$ is utilized, and $\psi_{N,\tilde{n}}\left(x_{N}\right)$ in position basis of the *N*th plane is calculated by putting j=N and $n=\tilde{n}$ in the defined wave function $\psi_{j,n}\left(x_{j}\right)$ in Equation (Equation 5). Similarly, inner product between history states is defined as follows:(23)$$\begin{array}{c} \begin{array}{c} \left(\psi_{N,\hat{n}}|\psi_{N,\tilde{n}}\right)=\int_{x_{N}=-\infty }^{\infty }dx_{N}\;\psi_{N,\hat{n}}^{*}\left(x_{N}\right)\;\psi_{N,\tilde{n}}\left(x_{N}\right) \end{array} \end{array}$$
The probability for the light to be diffracted through the *i*th slit on the *j*th plane with the projection $P_{j,i}$ is denoted by $Prob_{j,i}^{P}$. Similarly, probability to be measured on the *j*th plane with measurement projection $M_{j}$ is denoted by $Prob_{j}^{M}$. $Prob_{j,i}^{P}$ is calculated by using the weight of the compound history $\Omega_{N,\{j,i\}}$ including the targeted event $P_{j,i}$ as follows:(24)$$\begin{array}{cc} Prob_{j,i}^{P}\equiv & \left(\Omega_{N,\{j,i\}}|\Omega_{N,\{j,i\}}\right) \end{array}$$
where $\Omega_{N,\{j,i\}}$ is defined as follows: (25)$$\begin{array}{c} \begin{array}{c} \Omega_{N,\{j,i\}}=\sum_{n}\sum_{i_{j-1}=1}^{S_{j-1}}\sum_{i_{j-2}=1}^{S_{j-2}}\ldots \sum_{i_{1}=1}^{S_{1}}\;\left(\left[O_{n}\left(t_{N}\right)\right]⊙\ldots ⊙\left[O_{n}\left(t_{j+1}\right)\right]⊙\left[P_{j,i}\right]⊙\left[P_{j-1,i_{j-1}}\right]⊙\ldots ⊙\left[P_{1,i_{1}}\right]⊙\left[\rho_{0}\right]\right) \end{array} \end{array}$$
where elementary diffraction history states include diffraction events $P_{l,i_{l}}$ for l<j and $i_{l}\in \left[1,S_{l}\right]$ until the *j*th plane and diffraction event $\left[P_{j,i}\right]$ on the *j*th plane at $t_{j}$, and where the events $\left[O_{n}\left(t_{j+1}\right)\right]$ to $\left[O_{n}\left(t_{N}\right)\right]$ denote any dynamically possible projector at the times between $t_{j+1}$ and $t_{N}$. Probability for the events after diffraction will not have any effect on diffraction probability through $P_{j,i}$, and those projections are discarded. Then, it is easily calculated by using Equations (Equation 5) and (Equation 7) and with $\langle \Psi_{j}|\;P_{j,i}^{†}\;P_{j,i}\;|\Psi_{j}\rangle$ as follows: (26)$$\begin{array}{c} Prob_{j,i}^{P}=\int_{-\infty }^{\infty }dx_{j}\;e^{-\frac{{\left(x_{j}-X_{j,i}\right)}^{2}}{2\;\beta_{j,i}^{2}}}\;|\Psi_{j}\left(x_{j}\right){|}^{2} \end{array}$$
$Prob_{j}^{M}$ is calculated with $Prob_{j}^{M}=\int_{-\infty }^{\infty }dx_{j}\;|m_{j}\left(x_{j}\right)\Psi_{j}\left(x_{j}\right){|}^{2}$. Similarly, diffraction through one of several slits in a superposition of *s* slits on the *j*th plane is given by the following expression: (27)$$\begin{array}{c} Prob_{j,{\tilde{i}}_{s}}^{P}=\int_{-\infty }^{\infty }dx_{j}\;{\left(\sum_{i\in {\tilde{i}}_{s}}e^{-\frac{{\left(x_{j}-X_{j,i}\right)}^{2}}{2\beta_{j,i}^{2}}}\right)}^{2}\;|\Psi_{j}\left(x_{j}\right){|}^{2} \end{array}$$
where ${\tilde{i}}_{s}\equiv \left\{i_{1},i_{2},\ldots ,i_{s}\right\}$ and $i_{l}\in \left[1,S_{j}\right]$ for l∈[1,s], $i_{a}\neq i_{b}$.

It is important to emphasize the practical meaning of the probabilities of the light diffractions or measurements on the plane. In practice, the probabilities are proportional to the number of photons for each event, e.g., the number of photons passing through a particular slit integrated over a long measurement time for calculating projection probabilities or the number of photons detected on the specific area of the planes for the measurement projection. Histogram-based modeling for counting the photons for all planes and the slits provides the normalized overall probability for each event summing to a total of unity. Photon or particle counting with classical light is already achieved in various studies characterizing the exotic properties of the paths [31] or Fresnel diffraction properties [30].

Quantumness and temporal correlations for the MPD system design are analyzed by explicitly providing theoretical formulation of LGIs next.

### 2.4. Modeling of the Violation of LGI in MPD

LGIs test the temporal correlations by measuring at different times in analogy with spatial Bell’s inequalities for the entanglement between spatially separated systems [12,13]. Three-time correlation-based inequality is defined in Equation (Equation 2) as $K=C_{01}+C_{12}-C_{02}\leq 1$, where the bound is violated quantum mechanically with dichotomic systems, i.e., $Q_{j}=\pm 1$ for j∈[0,2], reaching the bound 3/2 for a two-level system with the maximum LGI violation of 1/2. $C_{j_{1},j_{2}}\equiv \langle Q_{j_{1}}Q_{j_{2}}\rangle$ is the expected value of the multiplication of the dichotomic observables, which is equal to $C_{j_{1},j_{2}}=\sum_{j_{1},j_{2}}p\left(j_{1},j_{2}\right)Q_{j_{1}}Q_{j_{2}}$, where $p(j_{1},j_{2})$ is the probability for the measurement of $Q_{j_{1}}$ and $Q_{j_{2}}$ at times $t_{j_{1}}$ and $t_{j_{2}}$, respectively, and $t_{2}>t_{1}>t_{0}$. Noninvasiveness or non-disturbing structure of the measurements should be clearly satisfied in order to reduce the “clumsiness loophole” [15,16], i.e., experimental limitations and disturbance of the clumsy measurements making it difficult to convince a macrorealist. In Reference [16], ambiguous measurements are utilized to revise Equation (Equation 2) by including the effect of signaling. In this article, the same formulation is extended for the MPD setup exploiting simple architecture of slits.

The correlation and entanglement in time are tested with the two-plane setup, where each plane includes triple slits as shown in Figure 2a. It is assumed that the light diffracting through the first plane is taken into account while calculating probability amplitudes, i.e., utilizing negative measurement techniques. For example, if the measured state is set to $P_{1,1}$, then the second and third slits are closed, forcing the light to diffract through only the first slit setting the measurement result. Furthermore, denote $p_{1}\left(i_{1,1}\right)\equiv Prob_{1,i_{1,1}}^{P}$ and $p_{1}\left(\left\{i_{1,1},i_{1,2}\right\}\right)\equiv Prob_{1,i_{s}}^{P}$, where $i_{s}=\left\{i_{1,1},i_{1,2}\right\}$ for $i_{1,1},i_{1,2}\in \left[1,3\right]$ and $i_{1,1}\neq i_{1,2}$. The probability $p_{1}\left(\left\{i_{1,1},i_{1,2}\right\}\right)$ corresponds to the measurement result for $P_{1,i_{1,1}}\cup P_{1,i_{1,2}}$ being projected in one of the slits with the indices $i_{1,1}$ and $i_{1,2}$ on the first plane. Similarly, $p_{1}\left(\left\{1,2,3\right\}\right)$ denotes the overall projection on superposition in all three slits. On the other hand, assume that $p_{1,2}\left(\left\{i_{1,1},i_{1,2}\right\},{\hat{i}}_{2}\right)$ denotes the probability for the history:(28)$$\begin{array}{c} \begin{array}{c} \left[O_{{\hat{i}}_{2}}\left(t_{2}\right)\right]⊙\left(\left[P_{1,i_{1,1}}\right]+\left[P_{1,i_{1,2}}\right]\right)⊙\left[\rho_{0}\right] \end{array} \end{array}$$
where $\left[O_{{\hat{i}}_{2}}\left(t_{2}\right)\right]$ is one of $\left[P_{2,1}\right]$, $\left[P_{2,2}\right]$, $\left[P_{2,3}\right]$, or $\left[M_{2}\right]$ denoted by ${\hat{i}}_{2}=1,2,3$, and 4, respectively. Similar to Equations (Equation 26) and (Equation 27), $p_{1,2}\left(\left\{i_{1,1},i_{1,2}\right\},{\hat{i}}_{2}\right)$ for ${\hat{i}}_{2}\in \left[1,3\right]$ is found as follows: (29)$$\begin{array}{c} p_{1,2}\left(\left\{i_{1,1},i_{1,2}\right\},{\hat{i}}_{2}\right)=\int_{-\infty }^{\infty }dx_{2}\;{\left(\exp\left(-{\left(x_{2}-X_{2,{\hat{i}}_{2}}\right)}^{2}\;/\;\left(2\;\beta_{2,{\hat{i}}_{2}}^{2}\right)\right)\right)}^{2}\;|\psi_{2,i_{1,1}}\left(x_{2}\right)+\psi_{2,i_{1,2}}\left(x_{2}\right){|}^{2} \end{array}$$
where the elementary wave function is found with Equation (Equation 5) by using *i* as the path index as follows:(30)$$\begin{array}{c} \begin{array}{c} \psi_{2,i}\left(x_{2}\right)=\chi_{0}\sqrt{\xi_{1}}e^{\Gamma_{1}\;x_{2}^{2}}\;e^{H_{1}\;X_{1,i}^{2}}\;e^{r_{1}\;X_{1,i}\;x_{2}} \end{array} \end{array}$$
where $\Gamma_{1}=A_{1}\;+\;ı\;B_{1}$, $H_{1}=H_{R,1}\;+\;ı\;H_{I,1}$, $r_{1}=c_{1}\;+\;ı\;d_{1}$, i∈[1,3], $\chi_{0}$, $\xi_{1}$, $A_{1}$, $B_{1}$, $H_{R,1}$, $H_{I,1}$, $c_{1}$, and $d_{1}$ are defined in Section 4. Similarly, $p_{1,2}\left(\left\{i_{1,1},i_{1,2}\right\},4\right)$ is defined as follows: (31)$$\begin{array}{c} p_{1,2}\left(\left\{i_{1,1},i_{1,2}\right\},4\right)=\int_{-\infty }^{\infty }dx_{2}\;|m_{2}\left(x_{2}\right){|}^{2}\;|\psi_{2,i_{1,1}}\left(x_{2}\right)+\psi_{2,i_{1,2}}\left(x_{2}\right){|}^{2} \end{array}$$
where $i_{1,1},i_{1,2}\in \left[1,3\right]$ and $i_{1,1}\neq i_{1,2}$. $p_{1,2}\left(i_{1,1},{\hat{i}}_{2}\right)$, and $p_{1,2}\left(\left\{1,2,3\right\},{\hat{i}}_{2}\right)$ denote the probabilities for $\left[O_{{\hat{i}}_{2}}\left(t_{2}\right)\right]⊙\left[P_{1,i_{1,1}}\right]⊙\left[\rho_{0}\right]$ and $\left[O_{{\hat{i}}_{2}}\left(t_{2}\right)\right]⊙\left(\left[P_{1,1}\right]+\left[P_{1,2}\right]+\left[P_{1,3}\right]\right)⊙\left[\rho_{0}\right]$, where $i_{1,1}\in \left[1,3\right]$ and ${\hat{i}}_{2}\in \left[1,4\right]$. The same formulation is valid also for the second plane for $p_{2}\left(i_{2,1}\right)$, $p_{2}\left(\left\{i_{2,1},i_{2,2}\right\}\right)$, and $p_{2}\left(\left\{1,2,3\right\}\right)$ for $i_{2,1},i_{2,2}\in \left[1,3\right]$ and $i_{2,1}\neq i_{2,2}$. Observe that, at time $t_{2}$, it is assumed that $M_{2}$ is also included in calculations providing a complete set $I=M_{2}+\sum_{i=1}^{3}P_{2,i}$. Negative measurement methodology for the first plane is utilized such that the light only diffracting through the first plane is utilized in calculating probabilities. Therefore, all the probability calculations based on Equations (Equation 26), (Equation 27), (Equation 29), and (Equation 31) are normalized by $\Gamma_{c}\equiv (\sum_{i=1}^{3}Prob_{1,i}^{P}{)}^{-1}$. The probabilities denoted by $p_{j}\left(i_{j,1}\right)$, $p_{j}\left(\left\{i_{j,1},i_{j,2}\right\}\right)$, $p_{j}\left(\left\{1,2,3\right\}\right)$ for j∈[1,2], $p_{1,2}\left(i_{1,1},{\hat{i}}_{2}\right)$, $p_{1,2}\left(\left\{i_{1,1},i_{1,2}\right\},{\hat{i}}_{2}\right)$, and $p_{1,2}\left(\left\{1,2,3\right\},{\hat{i}}_{2}\right)$ are assumed to be normalized through the rest of the article. The normalized operator is defined as $P_{1,i}^{N}\equiv \Gamma_{c}\;P_{1,i}$ for i∈[1,3].

Assume that an ambiguous measurement set of three projections composed by $\left[O_{1}^{A}\left(t_{1}\right)\right]\equiv \left[P_{1,1}^{N}\right]+\left[P_{1,2}^{N}\right]$, $\left[O_{2}^{A}\left(t_{1}\right)\right]\equiv \left[P_{1,1}^{N}\right]+\left[P_{1,3}^{N}\right]$, and $\left[O_{3}^{A}\left(t_{1}\right)\right]\equiv \left[P_{1,2}^{N}\right]+\left[P_{1,3}^{N}\right]$ is defined. The setups for ambiguous measurements are shown in Figure 2b–d, respectively. In addition, an assignment of dichotomic indices for the measurement results is designed denoted by $Q_{1,i_{1}^{A}}\equiv \pm 1$ and $Q_{2,{\hat{i}}_{2}}\equiv \pm 1$ for $\left[O_{i_{1}^{A}}^{A}\left(t_{1}\right)\right]$ and $\left[O_{{\hat{i}}_{2}}\left(t_{2}\right)\right]$, respectively, where $i_{1}^{A}\in \left[1,3\right]$ and ${\hat{i}}_{2}\in \left[1,4\right]$. $Q_{0}\equiv 1$ denotes initial condition $\left[\rho_{0}\right]$ with unity probability. These dichotomic indices can be assigned arbitrarily while they are chosen in Section 2.6 based on the maximization of LGI violation by comparing all the possible assignment combinations. Then, utilizing a similar architecture to the ambiguous LGI, i.e., Equation (Equation 14) in Reference [16], a conversion matrix D is defined inferring the probability $p_{1}\left(i_{1,1}\right)$ from the ambiguous measurements with ${\hat{p}}_{1}\left(i_{1,1}\right)\equiv \sum_{i_{1}^{A}}D_{i_{1,1},i_{1}^{A}}\;p_{1}^{A}\left(i_{1}^{A}\right)$, where $p_{1}^{A}\left(i_{1}^{A}\right)$ denotes the probability for the history $\left[O_{i_{1}^{A}}^{A}\left(t_{1}\right)\right]⊙\left[\rho_{0}\right]$ for $i_{1}^{A}\in \left[1,3\right]$, $D_{i_{1,1},i_{1}^{A}}$ is the element at the $i_{1,1}$th row and the $i_{1}^{A}$th column of the conversion matrix D, and ${\hat{p}}_{1}\left(i_{1,1}\right)$ denotes the inferred probability such that a macrorealist will not observe any problem. Similarly, ${\hat{p}}_{1,2}\left(i_{1,1},{\hat{i}}_{2}\right)$ becomes the following:(32)$$\begin{array}{c} {\hat{p}}_{1,2}\left(i_{1,1},{\hat{i}}_{2}\right)\equiv \sum_{i_{1}^{A}}D_{i_{1,1},i_{1}^{A}}\;p_{1,2}^{A}\left(i_{1}^{A},{\hat{i}}_{2}\right)=\sum_{i_{1}^{A}}D_{i_{1,1},i_{1}^{A}}\;\Gamma_{c}\;p_{1,2}\left(\left\{i_{1,1}^{A},i_{1,2}^{A}\right\},{\hat{i}}_{2}\right) \end{array}$$
where $p_{1,2}^{A}\left(i_{1}^{A},{\hat{i}}_{2}\right)$ denotes the probability for the history $\left[O_{{\hat{i}}_{2}}\left(t_{2}\right)\right]⊙\left[O_{i_{1}^{A}}^{A}\left(t_{1}\right)\right]⊙\left[\rho_{0}\right]$, where $i_{1,1},i_{1}^{A}\in \left[1,3\right]$ and ${\hat{i}}_{2}\in \left[1,4\right]$, while it is found in Equations (Equation 29) and (Equation 31) by normalizing as follows:(33)$$p_{1,2}^{A}\left(i_{1}^{A},{\hat{i}}_{2}\right)\equiv \Gamma_{c}\;p_{1,2}\left(\left\{i_{1,1}^{A},i_{1,2}^{A}\right\},{\hat{i}}_{2}\right)$$
where $i_{1,1}^{A}$ and $i_{1,2}^{A}$ correspond to the the event $\left[O_{i_{1}^{A}}^{A}\left(t_{1}\right)\right]$, i.e., $\left\{i_{1,1}^{A},i_{1,2}^{A}\right\}$ equals {1,2}, {1,3}, and {2,3} for $i_{1}^{A}\equiv 1$, 2, and 3, respectively. For example, for the proposed setup, ${\hat{p}}_{1}\left(1\right)=\left(p_{1}^{A}\left(1\right)+p_{1}^{A}\left(2\right)-p_{1}^{A}\left(3\right)\right)\;/\;2$ since the following probability relation holds:(34)$$Prob_{1,1}^{P}=\frac{Prob_{1,\{1,2\}}^{P}+Prob_{1,\{1,3\}}^{P}-Prob_{1,\{2,3\}}^{P}}{2}$$
Therefore, $D_{11}=0.5$, $D_{12}=0.5$, and $D_{13}=-0.5$. Similarly, $D_{21}=D_{23}=D_{32}=D_{33}=0.5$ and $D_{22}=D_{31}=-0.5$. Then, a macrorealist is convinced that the inferred probabilities are utilized for the calculations of $C_{01}$, $C_{12}$, and $C_{02}$ with Equation (Equation 2) by replacing $p_{1,2}\left(i_{1,1},{\hat{i}}_{2}\right)$ with ${\hat{p}}_{1,2}\left(i_{1,1},{\hat{i}}_{2}\right)$ and by properly defining the degree of signaling level between the first and second planes for the ambiguous measurements increasing the required LGI bound. Then, following the similar methodology in Reference [16] (Equations (5) and (14) in Reference [16]), $K=C_{01}+C_{12}-C_{02}$ with $Q_{0}=1$ is easily transformed into the combination of the standard LGI term free of the invasive measurement and a signaling term as follows by firstly replacing the measured probabilities with the inferred ones and then by inserting into *K*:(35)$$\begin{array}{c} \begin{array}{cc} K_{A}\equiv & \sum_{i_{1}^{A}=1}^{3}\sum_{i_{1,1}=1}^{3}\sum_{{\hat{i}}_{2}=1}^{4}\left(Q_{1,i_{1,1}}\;+\;Q_{1,i_{1,1}}Q_{2,{\hat{i}}_{2}}\;-Q_{2,{\hat{i}}_{2}}\;\right)D_{i_{1,1},i_{1}^{A}}\;p_{1,2}^{A}\left(i_{1}^{A},{\hat{i}}_{2}\right)\;-\;\sum_{{\hat{i}}_{2}=1}^{4}Q_{2,{\hat{i}}_{2}}\Delta_{S}\left({\hat{i}}_{2}\right) \end{array} \end{array}$$
where the first term is the standard LGI definition with inferred probabilities and the second term includes inferred signaling terms $\Delta_{S}\left({\hat{i}}_{2}\right)$ between the first and second planes for the measurement $\left[O_{{\hat{i}}_{2}}\left(t_{2}\right)\right]$ for each ${\hat{i}}_{2}$. It is modeled as a signaling quantifier showing the influence of the measurement at time $t_{1}$ to the measurement at time $t_{2}$ and is defined by utilizing ambiguous measurements as follows:(36)$$\begin{array}{ccc} \Delta_{S}\left({\hat{i}}_{2}\right)\;\equiv \;p_{2}\left({\hat{i}}_{2}\right)\;-\;\sum_{i_{1,1}=1}^{3}{\hat{p}}_{1,2}\left(i_{1,1},{\hat{i}}_{2}\right)\; & = & \;p_{2}\left({\hat{i}}_{2}\right)\;-\;\sum_{i_{1,1}=1}^{3}\sum_{i_{1}^{A}=1}^{3}D_{i_{1,1},i_{1}^{A}}\;p_{1,2}^{A}\left(i_{1}^{A},{\hat{i}}_{2}\right)\; \end{array}$$(37)$$\begin{array}{ccc} & = & \;p_{2}\left({\hat{i}}_{2}\right)\;-\;\frac{1}{2}\sum_{i_{1}^{A}=1}^{3}p_{1,2}^{A}\left(i_{1}^{A},{\hat{i}}_{2}\right) \end{array}$$
where $\left(\sum_{i_{1,1}=1}^{3}D_{i_{1,1},i_{1}^{A}}\right)=1\;/\;2$. Therefore, the no-signaling-in-time (NSIT) condition for the definition with ambiguous measurement of $\Delta_{S}\left({\hat{i}}_{2}\right)=0$ is expected to convince a macrorealist about the reliability of the measurement setup. Then, the violation of LGI free of the invasive measurement becomes the following:(38)$$K_{A}\leq K_{V}\equiv 1+\sum_{{\hat{i}}_{2}=1}^{4}\left|\Delta_{S}\left({\hat{i}}_{2}\right)\right|$$
where the summation at the right side of the inequality, i.e., $K_{V}-1$, shows the invasiveness of the measurements and the signaling level. Therefore, measured values $p_{1,2}^{A}\left(i_{1}^{A},{\hat{i}}_{2}\right)$ are utilized to check the violation compatible with respect to the objections of a macrorealist. If Equations (Equation 29), (Equation 31), and (Equation 33) are inserted into Equations (Equation 35), (Equation 36), and (Equation 38), then the following is obtained: (39)$$\begin{array}{ccc} K_{A} & = & G_{1}\;\sum_{{\hat{i}}_{2}=1}^{3}Q_{2,{\hat{i}}_{2}}f_{2}\left(X_{2,{\hat{i}}_{2}}\right)\;-\;G_{1}\;Q_{2,4}\sum_{{\hat{i}}_{2}=1}^{3}f_{2}\left(X_{2,{\hat{i}}_{2}}\right)\;+\;G_{1}\;\sum_{i_{1,1}=1}^{3}Q_{1,i_{1,1}}\sum_{{\hat{i}}_{2}=1}^{3}Q_{2,{\hat{i}}_{2}}f_{1,2}\left(X_{1,i_{1,1}},X_{2,{\hat{i}}_{2}},{\vec{l}}_{i_{1,1}}\right)\\ & & -\;G_{1}\;Q_{2,4}\sum_{i_{1,1}=1}^{3}Q_{1,i_{1,1}}\sum_{{\hat{i}}_{2}=1}^{3}f_{1,2}\left(X_{1,i_{1,1}},X_{2,{\hat{i}}_{2}},{\vec{l}}_{i_{1,1}}\right)\;+\;G_{2}\;\left(1+Q_{2,4}\right)\;\sum_{i_{1,1}=1}^{3}Q_{1,i_{1,1}}f_{1}\left(X_{1,i_{1,1}},{\vec{l}}_{i_{1,1}}\right)\\ & & -\;G_{2}\;Q_{2,4}\sum_{i_{1,1}=1}^{3}f_{1}\left(X_{1,i_{1,1}},{\vec{l}}_{i_{1,1}}\right)\;-\;\;G_{2}\;Q_{2,4}\;f_{T}\;\;\;\;\;\;\;\;\;\;\; \end{array}$$
(40)$$\begin{array}{ccc} K_{V} & = & 1+\sum_{{\hat{i}}_{2}=1}^{3}\left|f_{V}\left(X_{2,{\hat{i}}_{2}}\right)\right|+|f_{T}\;G_{2}\;-\sum_{{\hat{i}}_{2}=1}^{3}f_{V}\left(X_{2,{\hat{i}}_{2}}\right)| \end{array}$$
where ${\vec{l}}_{1}={\left[1\;\;1\;\;-1\right]}^{T}$; ${\vec{l}}_{2}={\left[1\;\;-1\;\;1\right]}^{T}$; ${\vec{l}}_{3}={\left[-1\;\;1\;\;1\right]}^{T}$; the functions $f_{1}\left(.\right)$, $f_{2}\left(.\right)$, $f_{1,2}\left(.\right)$, and $f_{V}\left(.\right)$; and the variables $f_{T}$, $G_{1}$, and $G_{2}$ are defined in Section 4. Next, quantum interference among the paths, i.e., QPI, is defined for the MPD setup.

### 2.5. Modeling of QPI

Double-slit interference gives a clear indication of quantumness showing wave-particle duality and spatial interference as emphasized by Feynman. MPD setup presents the complementary phenomenon of the temporal interference among the paths which cannot be explained in any classical way showing that paths interfere in time, destructively and constructively decreasing and increasing the probability of the consecutive events, respectively. A *gedanken* experiment shown in Figure 3 is designed with three planes. The target is to analyze interference effects of opening both slits on the first plane in terms of the probability of the light to diffract through first (PL-1), second (PL-2), and third (PL-3) planes. History states at times $t_{1}$, $t_{2}$, and $t_{3}$ with three types of projections indexed by the superscripts *a*, *b*, and *c* are defined with the setups shown in Figure 3a–c, respectively, as follows:(41)$$\begin{array}{ccc} |\Psi_{1}^{a})\; & \equiv & \;P_{s}⊙\left[\rho_{0}\right] \end{array}$$(42)$$\begin{array}{ccc} |\Psi_{2}^{a})\; & \equiv & \;\left[P_{2,1}\right]⊙P_{s}⊙\left[\rho_{0}\right]; \end{array}$$(43)$$\begin{array}{ccc} |\Psi_{3}^{a})\; & \equiv & \;\left[P_{3,1}\right]⊙\left[P_{2,1}\right]⊙P_{s}⊙\left[\rho_{0}\right] \end{array}$$(44)$$\begin{array}{ccc} |\Psi_{1}^{b})\; & \equiv & \;\left[P_{1,1}\right]⊙\left[\rho_{0}\right] \end{array}$$(45)$$\begin{array}{ccc} |\Psi_{2}^{b})\; & \equiv & \;\left[P_{2,1}\right]⊙\left[P_{1,1}\right]⊙\left[\rho_{0}\right] \end{array}$$(46)$$\begin{array}{ccc} |\Psi_{3}^{b})\; & \equiv & \;\left[P_{3,1}\right]⊙\left[P_{2,1}\right]⊙\left[P_{1,1}\right]⊙\left[\rho_{0}\right] \end{array}$$(47)$$\begin{array}{ccc} |\Psi_{1}^{c})\; & \equiv & \;\left[P_{1,2}\right]⊙\left[\rho_{0}\right] \end{array}$$(48)$$\begin{array}{ccc} |\Psi_{2}^{c})\; & \equiv & \;\left[P_{2,1}\right]⊙\left[P_{1,2}\right]⊙\left[\rho_{0}\right] \end{array}$$(49)$$\begin{array}{ccc} |\Psi_{3}^{c})\; & \equiv & \;\left[P_{3,1}\right]⊙\left[P_{2,1}\right]⊙\left[P_{1,2}\right]⊙\left[\rho_{0}\right] \end{array}$$
where superposition event at time $t_{1}$ is defined as $P_{s}\equiv \left[P_{1,1}\right]+\left[P_{1,2}\right]$ and the event probabilities are defined as follows:(50)$$\begin{array}{c} \;p_{1}\left(\left\{1,2\right\}\right)\;\equiv \;\left(\Psi_{1}^{a}|\Psi_{1}^{a}\right);\;p_{1}\left(1\right)\;\equiv \;\left(\Psi_{1}^{b}|\Psi_{1}^{b}\right);\;p_{1}\left(2\right)\;\equiv \;\left(\Psi_{1}^{c}|\Psi_{1}^{c}\right) \end{array}$$(51)$$\begin{array}{c} \;p_{1,2}\left(\left\{1,2\right\},1\right)\;\equiv \;\left(\Psi_{2}^{a}|\Psi_{2}^{a}\right);\;p_{1,2}\left(1,1\right)\;\equiv \;\left(\Psi_{2}^{b}|\Psi_{2}^{b}\right);\;p_{1,2}\left(2,1\right)\;\equiv \;\left(\Psi_{2}^{c}|\Psi_{2}^{c}\right) \end{array}$$(52)$$\begin{array}{c} p_{1,2,3}\left(\left\{1,2\right\},1,1\right)\;\equiv \;\left(\Psi_{3}^{a}|\Psi_{3}^{a}\right);\;p_{1,2,3}\left(1,1,1\right)\;\equiv \;\left(\Psi_{3}^{b}|\Psi_{3}^{b}\right);\;p_{1,2,3}\left(2,1,1\right)\;\equiv \;\left(\Psi_{3}^{c}|\Psi_{3}^{c}\right) \end{array}$$
It is observed that $p_{1}\left(\left\{1,2\right\}\right)=p_{1}\left(1\right)+p_{1}\left(2\right)$ while interference exists among consecutive planes. The targeted scenario for the relation between $|\Psi_{3}^{a})$, i.e., the superposition of $|\Psi_{3}^{b})$ and $|\Psi_{3}^{c})$, and $|\Psi_{3}^{c})$ is as follows: (53)$$\begin{array}{ccc} p_{1}\left(\left\{1,2\right\}\right) & \;\overset{1}{>}\; & p_{1}\left(2\right) \end{array}$$(54)$$\begin{array}{ccc} p_{1,2}\left(\left\{1,2\right\},1\right) & \;\overset{2}{>}\; & p_{1,2}\left(2,1\right) \end{array}$$(55)$$\begin{array}{ccc} p_{1,2,3}\left(\left\{1,2\right\},1,1\right) & \;\overset{3}{<}\; & p_{1,2,3}\left(2,1,1\right) \end{array}$$
The superposition of $|\Psi_{3}^{b})$ and $|\Psi_{3}^{c})$ on PL-1 increases the probability for the light to diffract at time $t_{1}$ in $\overset{1}{>}$. In $\overset{2}{>}$, the superposition constructively interferes to increase also the probability to diffract through the slit on PL-2 at time $t_{2}$. However, they destructively interfere in $\overset{3}{<}$, decreasing the probability to diffract through the slit on PL-3 at time $t_{3}$. Assuming starting with the setup in Figure 3c, with the second slit (the one with $X_{1,2}>0$) open on PL-1, if the first slit is additionally opened as shown in Figure 3a, then the probability for the light to diffract through PL-1 and PL-2 increases while decreasing the probability to diffract through PL-3. The counterintuitive observation based on a classical logic with balls passing through the slits is described as follows. We open the second slit on PL-1 and observe that the total number of balls passed through them for a statistical experiment increases. It becomes more probable to pass through PL-1 with two slits in a classically logical manner. Furthermore, the probability to pass through the single slit on PL-2 or the the total number of balls passing through PL-2 somehow increases. However, we observe that the probability to pass through the single slit on the consecutive PL-3 counterintuitively decreases in spite of the fact that more balls are coming from the second plane. This is complementary to the conventional spatial interference extensively studied in double-slit interference experiments.

QPI modeling requires the calculation of $\psi_{3,j}\left(x_{3}\right)$ in addition to $\psi_{2,j}\left(x_{3}\right)$ for j∈[0,1] as modeled in Equation (Equation 30). The explicit parameters required for the calculation of both $\psi_{2,j}\left(x_{3}\right)$ and $\psi_{3,j}\left(x_{3}\right)$ are presented in Section 4.

### 2.6. Numerical Results

Two different calculation scenarios are performed denoted by $Sim_{1}$ and $Sim_{2}$ as shown in Table 1. $Sim_{1}$ calculates violation of LGI while $Sim_{2}$ shows an example of interference in time for the numerical analysis of QPI. Physical parameters are monochromatic laser source wavelength λ=650 (nm) as a widely available resource, light velocity of $3\;\times \;10^{8}$ (m/s) in the z-direction, and $\hbar =1.05\;\times \;10^{-34}$ (J × s) as Planck’s constant. The wavelength allows another degree of freedom to be adapted based on experimental design requirements or the targeted system design. The layouts used in the simulations $Sim_{1}$ and $Sim_{2}$ are shown in Figure 4a,b, respectively. Furthermore, illustrative measurement setups for practically counting the number of photons compared with the emitted photons in unit time are presented in Figure 4c,d in order to calculate the probabilities $p_{1}\left(\left\{1,2\right\}\right)$ and $p_{1,2}\left(\left\{1,3\right\},2\right)$, respectively.

#### 2.6.1. Violation of LGI

There are two planes (PL-1 and PL-2) as shown in Figure 2 and Figure 4a. The slit positions on PL-1 and PL-2 are set to $D_{s}\;+\;\left[-\Delta x\;\;0\;\;\Delta x\right]\times \beta_{1}$ and $\left[-\Delta x\;\;0\;\;\Delta x\right]\times \beta_{2}$, respectively, where the positions on PL-1 are shifted with varying $D_{s}$ and the ratio of inter-slit distances to the slit widths is fixed in both planes. Δx is chosen larger than seven to realize independence of Gaussian slits, i.e., $\exp({\left(X_{1,i}-X_{1,i+1}\right)}^{2}\;/\;2\;\beta_{1}^{2})\ll 1$. The distance between the planes is set to *L* such that the time duration is $t_{01}=t_{12}\equiv L\;/\;c$. Gaussian source parameter $\sigma_{0}$ is varied between 10 (μm) and 800 (μm), compatible with standard laser resources including fiber lasers allowing smaller diameters reaching tens of micrometers.

The shift of the slits on PL-1 results in varying levels of temporal correlation for the diffraction through the slits on PL-2. LGI violation ($K_{A}\;-\;K_{V}$) is analyzed for varying $D_{s}$, $t_{01}=t_{02}$, $\beta_{1}$, $\beta_{2}$, and $\sigma_{0}$. In Figure 5a, it is shown with the signaling level ($K_{V}\;-\;1$) for varying $D_{s}$ for $t_{01}=0.2$ (ns), $t_{12}=0.1$ (ns), Δx=7, $\beta_{1}=15$ (μm), $\beta_{2}=30$ (μm), and $\sigma_{0}=130$ (μm). The maximum violation is analyzed for different values of $Q_{1,i_{1,1}}$ and $Q_{2,{\hat{i}}_{2}}$ for $i_{1,1}\in \left[1,3\right]$ and ${\hat{i}}_{2}\in \left[1,4\right]$, respectively, and the signs maximizing the violation are chosen for each $D_{s}$ shift. In Figure 5b, distributions of the sign assignments maximizing the violation are shown. Different setups realized with varying shift on PL-1 result in different optimized sign assignments for maximum violation. Furthermore, violation decreases as the interplane slit distance increases, i.e., decreasing to zero with $K_{A}=K_{V}\approx 1$. It is observed that LGI is violated significantly, reaching close to 0.3 for the specific setup shown in Figure 5a. The signaling $\sum_{k=1}^{4}\left|\Delta_{S}\left(k\right)\right|$ is close to zero, as shown in Figure 5a for the marked region between the violation peaks. In the next simulations, the signaling is shown to decrease approximately to zero for varying setup values. It is shown as a proof of concept in Figure 6a that the amount of signaling is $K_{V}-1\approx 2.3\times 10^{-3}$ for a violation of $K_{A}-K_{V}\approx 0.2122$ while satisfying NIM and signaling-in-time-related assumptions discussed in Reference [16] and utilizing a NIM-free violation.

In Figure 6, the effects of varying Δx, $\sigma_{0}$, and $t_{01}=t_{12}$ on the violation of LGI are shown for varying $\beta_{1}$ and $\beta_{2}$ pairs. $D_{s}$ and the signs of $Q_{1,i_{1,1}}$ and $Q_{2,{\hat{i}}_{2}}$ for $i_{1,1}\in \left[1,3\right]$ and ${\hat{i}}_{2}\in \left[1,4\right]$, respectively, are chosen to maximize the violation for each pair and specific $\sigma_{0}$ value. $\beta_{1}$ and $\beta_{2}$ are chosen in the sets {5,10,15,…,50} (μm) and {10,20,30,…,100} (μm), respectively, while choosing the maximum violation pairs. Similarly, $D_{s}$ is chosen in the interval of [1,1000] (μm) with the resolution of 1 (μm). There are important various observations for the specific simulation constraints in Table 1 which can be further improved by increasing the range of values and the resolution in simulations such as for the values of $\beta_{1}$, $\beta_{2}$, and $D_{s}$. However, the provided simple parameter set shows LGI violation under no-signaling conditions as a proof of concept. More specific observations provide more information about the nature of LGI violations for MPD setup as discussed next.

It is observed in Figure 6a that violation becomes smaller as the relative distance between slits compared with the slit width parameter β increases from Δx=7 to Δx=11 for $t_{01}=t_{12}=0.1$ (ns). Furthermore, $\sigma_{0}$ values maximizing the violation and the functional behavior with respect to varying $\sigma_{0}$ are approximately the same as Δx changes for fixed $t_{01}=t_{12}$. The range of violation reaches ≈0.4 and ≈0.23 for Δx=7 and Δx=11, respectively, while decreases as Δx increases. The signaling term is approximately zero for the Δx=11 case with the maximum violation amplitude of $K_{A}-K_{V}\approx 0.2122$ as emphasized previously. It is an open issue to design MPD setups maximizing the violation, e.g., similar to the boundary value of 0.5 observed in conventional QM setups or 0.464, as calculated in Reference [16] with inverted measurements for a three-level quantum optical system. Optimum slit width maximizing violation for each $\sigma_{0}$ decreases as Δx increases, as shown in Figure 6b. In other words, as Δx increases, $\beta_{1}$ and $\beta_{2}$ are getting smaller to stabilize $\Delta x\times \beta_{1}$ and $\Delta x\times \beta_{2}$ for better violation. As a result, increasing the relative inter-slit distance results in more classical behavior while decreasing violation. On the other hand, increasing $\sigma_{0}$ further does not have any effect on the optimum $\beta_{1}$, $\beta_{2}$, and $D_{s}$ as the source behaves as a plane wave for the specific setup. Violation amplitudes with respect to different $\beta_{1}$ and $\beta_{2}$ pairs for the maximizing $\sigma_{0}$ values (extracted from Figure 6a) of 30 (μm) and 150 (μm) for Δx=7 and Δx=11, respectively, are shown in Figure 6c,e, respectively, for $t_{01}=t_{12}=0.1$ (ns). There is a decrease in both the range of $\beta_{1}$ and $\beta_{2}$ values and the maximum violation for Δx=11. On the other hand, increasing interplane distance two times, i.e., making $t_{01}=t_{12}=0.2$ (ns), is observed not to change the maximum violation regime of 0.4 while increasing both the value of $\sigma_{0}$ for the maximum violation to 230 (μm), as shown in Figure 6a, and the ($\beta_{1}$, $\beta_{2}$) values giving the maximum violation amplitudes, as shown in Figure 6d. Increasing interplane distance improves the spread of the wave function on the consecutive plane while requires larger widths of source beam and slits in order to have similar violation amplitudes.

The results in Figure 6 assume fully coherent source both temporally and spatially. The realistic modeling of the temporal and spatial coherence of the light source is presented in Section 4.2. In our simulations, the total duration that light propagates is smaller than 1 ns, i.e., corresponding to $\approx 10^{-9}\times 3\times 10^{8}=30$ (cm), which is much smaller than the temporal coherence time of the conventional single-mode lasers, i.e., $\Delta t>10^{-6}$ (s) for single-mode fiber lasers with Δf of a few KHz [34]. On the other hand, the analysis of the spatial coherence is realized by defining the setup diameters $D_{1}$ and $D_{2}$ on the first and second planes, respectively, for the areas covering the slits. Then, these are compared with the spatial coherence diameter defined in Equation (Equation 58) depending on the duration of the propagation *t* from the source to the diffraction plane and on the standard deviation of the Gaussian source denoted by $\sigma_{0}$. The detailed modeling and discussion are provided in Section 4.2, where the spatial coherence diameters $D_{c}$ and diffraction setup diameters $D_{1}$ and $D_{2}$ are described and it is targeted that $D_{c}$ is larger than both $D_{1}$ and $D_{2}$ as described in Equations (Equation 59) and (Equation 60). Then, $D_{c}\left(t_{01},\sigma_{0}\right)$ corresponds to the propagation from the source to the first plane and $D_{c}\left(t_{12},\beta_{1}\right)$ is for the propagation from the first plane to the second plane. It is assumed that the Gaussian slit modifies the diffracted wave with $\beta_{1}$ as the new source standard deviation while leaving the analysis with respect to the parameters of the wave function in Equation (Equation 5), e.g., $A_{1}$, as an open issue, as discussed in Section 4.2. Assume that the analysis providing the minimum signaling with Δx=11 and $t_{01}=t_{12}=0.1$ (ns) is targeted for coherence. In Figure 6f, the comparisons of $D_{c}\left(t_{01},\sigma_{0}\right)$ vs. $D_{1}$ and $D_{c}\left(t_{12},\beta_{1}\right)$ vs. $D_{2}$ are shown. It is observed that the spatial coherence diameter covers the simulation parameters for both the peak and no-signaling cases. The LGI violation curve for Δx=11 in Figure 6a is plotted again in Figure 6g by also emphasizing the violation amplitudes for the simulation parameters in Figure 6f. It is an open issue to design MPD setups compatible with the coherence properties of the practical sources while satisfying quantum properties including the violation of LGI.

#### 2.6.2. QPI Analysis

The three-plane setup (PL-1, PL-2, and PL-3) shown in Figure 3 is numerically analyzed for fixed values of $\beta_{1}=25$ (μm), $\beta_{2}=35$ (μm), $\beta_{3}=45$ (μm), $\sigma_{0}=200$ (μm), $t_{01}=0.5$ (ns), $t_{12}\;=\;0.2$ (ns), $t_{23}\;=\;0.1$ (ns), and ${\vec{X}}_{1}^{T}=\left[-4\;\;\;\;4\right]\times \beta_{1}$. Sampling value of $T_{s}=1$ (μm) is utilized in the analysis. Constructive and destructive interferences of two different paths at times $t_{2}$ and $t_{3}$, respectively, are performed by designing the slit positions with respect to spatial constructive and destructive interferences on PL-2 and PL-3, respectively. In Figure 7a, single slit position for PL-2, i.e., $X_{2,1}$, is chosen on the constructive interference regions, where $|\psi_{2,0}\left(x_{2}\right)\;+\;\psi_{2,1}\left(x_{2}\right)|$ is larger than $|\psi_{2,1}\left(x_{2}\right)|$ due to the superposition. Then, for each constructive $X_{2,1}$ slit position, the destructive interference regions on PL-3 such that $|\psi_{3,0}\left(x_{3}\right)\;+\;\psi_{3,1}\left(x_{3}\right)|$ is smaller than $|\psi_{3,1}\left(x_{3}\right)|$ are searched while the magnitudes $\max_{x_{3}}\left\{|\psi_{3,0}\left(x_{3}\right)\;+\;\psi_{3,1}\left(x_{3}\right){|}^{2}-{\left|\psi_{3,1}\left(x_{3}\right)\right|}^{2}\right\}$ and the corresponding $X_{3,1}=x_{3}$ are shown in Figure 7b,c, respectively. It is observed that $X_{2,1}\approx 140\;\mu$m maximizes the destructive interference while the corresponding wave function amplitude on PL-3 is shown in Figure 7d, showing the the maximum destructive interference at $X_{3,1}\approx 143\;\mu$m. Then, for each $\left(X_{2,1},X_{3,1}\right)$ pair, probability amplitudes of the histories are shown in Figure 7e with the marked areas where constructive interference occurs on PL-2 but with the destructive interference obtained on PL-3. The conditions with counterintuitive nature in Equations (Equation 53)–(Equation 55) for interference in time are satisfied. The top two pairs of curves satisfy $p_{1}\left(\left\{1,2\right\}\right)\;>\;p_{1}\left(2\right)$ and $p_{1,2}\left(\left\{1,2\right\},1\right)\;>\;p_{1,2}\left(2,1\right)$ due to the constructive interference while the bottom pair of the curves satisfies $p_{1,2,3}\left(\left\{1,2\right\},1,1\right)\;<\;p_{1,2,3}\left(2,1,1\right)$. In other words, the probability for the light to diffract through the second plane is decreased after blocking the first slit on PL-1 which counterintuitively results in an increase for the diffraction probability through the third plane. As a result, the proposed design of the setup and utilization of spatial interference result in interference of the quantum paths in time for the projection histories with classically counterintuitive probabilistic results.

Besides that, similar to the analysis of spatial and temporal coherence properties of the violation of LGI, the parameter set resulting in the maximum constructive and destructive interferences on PL-2 and PL-3, respectively, is analyzed for compatibility with the coherence of practical sources. The inequalities defined in Equations (Equation 61)–(Equation 63) in Section 4.2 are targeted to be satisfied where the spatial coherence diameter $D_{c}$ and diffraction setup diameters $D_{1}$, $D_{2}$, and $D_{3}$ are described. The inequalities are calculated for the defined group of $\sigma_{0}$, $\beta_{1}$, $\beta_{2}$, $\beta_{3}$, $t_{01}$, $t_{12}$, and $t_{13}$, where $X_{3,1}$ is given in Figure 7c and $X_{2,1}\in \left[140,170\right]\;$ (μm) is targeted. It is found that the practical coherence properties are satisfied as shown in Figure 7f comparing $D_{c}\left(t_{12},\beta_{1}\right)$ with $D_{2}$ and $D_{c}\left(t_{23},\beta_{2}\right)$ with $D_{3}$ where $D_{c}\left(t_{01},\sigma_{0}\right)\approx 607$ (μm) $>D_{1}\approx 270$ (μm).

## 3. Discussion and Conclusions

In this article, two novel resources for quantum technologies, i.e., QPE and QPI, are introduced based on tensor product structure of quantum history states for the simple linear optical setup of MPD. Operator theory modeling is presented by combining conventional history-based approaches of Griffith [3,4,5] and entangled histories framework of Reference [1] with FPI modeling. The inherent Feynman path generation mechanism of MPD setup is exploited for realizing quantum trajectories and for studying quantum temporal correlations. Following the similar terminology of entangled quantum trajectories in previous formulations, the temporal correlation among the quantum propagation paths of MPD design is denoted with QPE as a novel quantum resource. The state of the light after diffraction through consecutive planes is represented as a superposition of different trajectories through the slits as the main definition of QPE. Its two fundamental properties are theoretically and numerically analyzed: violation of LGI as the temporal analog of Bell’s inequality with the ambiguous form and no-signaling recently proposed by Emary [16] and QPI as a counterintuitive phenomenon defining the quantum interference of histories. LGIs are violated reaching >0.2 with a NIM-free formulation complementing the recent experimental implementation of the violation of LGI in Reference [19] utilizing linear polarization degree of freedom of the classical light with a photon-counting simple MPD setup of classical light. Besides that, it is an open issue to further design novel structures allowing the implementation of specific QPE states in analogy with GHZ states experimentally implemented for entangled trajectories [11].

Furthermore, QPI is numerically analyzed for a scenario where the decrease in the number of photons diffracting through a plane counterintuitively results in an increase in the number of photons diffracting through the next plane due to interference between two quantum history or trajectory states. MPD design introducing QPE and QPI is providing a test bed to understand the nature of temporal correlations improving our understanding of quantum mechanics and to design novel structures exploiting history-based formulation for practical purposes of quantum technologies. The simplicity of both the setup and proposed theoretical modeling allows varying kinds of *gedanken* experiments for implementing paradoxes emphasizing the quantum nature with counterintuitive scenarios.

MPD has significantly low experimental complexity, relying on light sources of finite spatial and temporal coherence combined with conventional intensity detection or photon counting by promising near-future implementations. The experimental implementation is further simplified based on the mature science of Fourier optics with recent proposals in Reference [9]. Experimental and simulation studies for single-plane diffraction architectures verify the proposed theoretical modeling based on Fresnel diffraction of Fourier optics [30] and FPIs [31,35,36]. Therefore, proposed MPD theory simply extends previous modeling to multiple planes with FPI approach. One of the experimental challenges is the design of Gaussian slit, while any slit mask can be represented in terms of Gaussian basis as discussed in Reference [9]. Furthermore, based on the results in this article, any slit mask is potentially expected to have unique MPD setup parameters resulting in both the violation of LGI and QPI as a future work both to theoretically design and to experimentally verify. Moreover, these experimental implementations require only the calculation of total photon counts on the total area of the specific slits. This simplifies the slit design by allowing various geometries and the experimental setup regarding the detectors related to the size, position, precision, and noise [37]. The overall number of photon counts is satisfied more easily compared with detailed representations of the interference pattern in spatial basis.

QPE as an entanglement resource promises future applications for computing [8,9], communications [10], and varying quantum technologies while enriching the conventional resources of quantum entanglement based on correlations among multiple spatial quantum units. In addition, MPD-based formulation of QPE and QPI allows future studies emphasizing the importance of time in QM fundamentals such as regarding entanglement in time [7,38,39], quantum cosmology, and gravity [40,41,42].

## 4. Methods

### 4.1. Parameters for FPI Modeling of the Violation of LGI

LGI and wave functions are modeled with FPI modeling [8], and the resulting parameters to be utilized in Equations (Equation 5), (Equation 30), (Equation 39), and (Equation 40) are provided in Table 2. The functions utilized in Equations (Equation 39) and (Equation 40) are defined as follows, where the variables $k_{i}$ for i∈[1,11] defined in Table 2 depend on the system setup parameters $t_{01}$, $t_{12}$, $\beta_{1}$, $\beta_{2}$, and $\sigma_{0}$, where $f_{1}\left(x,\vec{l}\right)\;\equiv \;e^{k_{10}x^{2}}+{\vec{l}}^{T}\;{\vec{e}}_{4}$, $f_{2}\left(y\right)\;\equiv \;-2\;1_{3}^{T}\;{\vec{g}}_{y}-\sum_{i=1}^{3}e_{1}\left(X_{1,i},\;y\right)$, $f_{1,2}\left(x,y,\vec{l}\right)\;\equiv \;{\vec{l}}^{T}\;{\vec{g}}_{y}+\;e_{1}\left(x,y\right)$, $f_{V}\left(y\right)\;\equiv \;G_{1}\;1_{3}^{T}\;{\vec{g}}_{y}$, $f_{T}\;\equiv \;1_{3}^{T}\;{\vec{e}}_{4}$, $c\left(x_{1},x_{2},y\right)\;\equiv \;\cos\left(k_{1}\left(x_{1}^{2}\;-\;x_{2}^{2}\right)\;+\;k_{2}\;\left(x_{1}\;-\;x_{2}\right)\;y\right)$, $e_{1}\left(x,y\right)\;\equiv \;e^{-2\;k_{3}\;y\;x\;-\;k_{4}\;y^{2}\;+\;k_{5}\;x^{2}}$, $e_{2}\left(x_{1},x_{2},y\right)\;\equiv \;e^{k_{6}\;\left(x_{1}^{2}\;+\;x_{2}^{2}\right)\;-\;k_{3}\;\left(x_{1}\;+\;x_{2}\right)\;y-\;k_{4}\;y^{2}+k_{7}\;x_{1}\;x_{2}}$, $e_{4}\left(x_{1},x_{2}\right)\;\equiv \;e^{k_{8}\;\left(x_{1}^{2}\;+\;x_{2}^{2}\right)+k_{9}\;x_{1}\;x_{2}}$, ${\vec{e}}_{4}\;\equiv \;{\left[e_{4}\left(X_{1,1},X_{1,2}\right)\;\;e_{4}\left(X_{1,1},X_{1,3}\right)\;\;e_{4}\left(X_{1,2},X_{1,3}\right)\right]}^{T}$, $g_{i_{1},{\tilde{i}}_{1}}\left(y\right)\;\equiv \;c\left(X_{1,i_{1}},X_{1,{\tilde{i}}_{1}},\;y\right)\;e_{2}\left(X_{1,i_{1}},X_{1,{\tilde{i}}_{1}},\;y\right)$, ${\vec{g}}_{y}\;\equiv \;{\left[g_{1,2}\left(y\right)\;\;g_{1,3}\left(y\right)\;\;g_{2,3}\left(y\right)\right]}^{T}$, $G_{1}\;\equiv \;G_{2}\;\beta_{1}\;\beta_{2}\;m^{2}\;\sigma_{0}\;\sqrt{\left(\beta_{1}^{2}+d_{t,\sigma }\right)\;/\;k_{11}}$, $G_{2}\;\equiv \;1\;/\;\left(\sum_{i=1}^{3}e^{-X_{1,i}^{2}\;/\;\left(\beta_{1}^{2}+d_{t,\sigma }\right)}\right)$, $d_{t,\sigma }\;\equiv \;a_{t,\sigma }/\left(m^{2}\sigma_{0}^{2}\right)$, $\Xi_{i}\equiv \beta_{i}^{2}\;+\;\sigma_{0}^{2}$, $\vartheta_{t}\equiv m\;\sigma_{0}^{2}\;+\;ı\;\hbar \;t_{0,1}$, and $1_{3}={\left[1\;\;1\;\;1\right]}^{T}$. Besides that, the explicit parameters required for the calculation of $\psi_{3,i}\left(x_{3}\right)$ for i∈[0,1], i.e., two different paths through two different slits on the first plane and the single slit both on the second and the third planes, are shown in Table 3 based on the iterative modeling in Reference [8].

### 4.2. Temporal and Spatial Coherence of the Light Sources

The coherence of the light field is characterized both temporally and spatially [43]. Its temporal coherence length is inversely proportional to the the full width half maximum (FWHM) of the emission peak in the wavelength spectrum as follows [43,44]:(56)$$\Delta l\equiv \Delta t\;c=\sqrt{\frac{2\;ln2}{\pi \;n_{r}}}\frac{\lambda^{2}}{\Delta \lambda }$$
where $n_{r}$ is the refractive index of the medium and Δλ is the wavelength domain representation of the spectral width. The interference fringes in a double-slit experiment performed with this source are observed if the path length difference between different paths through the slits is smaller than Δl as described in Reference [43].

Spatial coherence depends on the size of the light field source where it is experimentally characterized with double-slit interference experiment as clearly described in Reference [43], as shown in Figure 8a. The source is assumed to have a side length Δs, and Δθ is the angle from the slits separated with $D_{c}$ to the center of the source. The distance between the source and slit planes is *L*. Then, spatial coherence distance $D_{c}$ satisfies the following relation for the interference fringes to be observed:(57)ΔθΔs≤λ
In Reference [45], more detailed calculation of the spatial coherence diameter of the 2D Gaussian intensity source $I\left(x_{0},y_{0}\right)=I_{0}\;\exp\left(-\left(x_{0}^{2}+y_{0}^{2}\right)\;/\;\left(2\;\sigma_{0}^{2}\right)\right)$ is achieved by using van Cittert–Zernike theorem [46]. It is observed that an incoherent source of uniform intensity with circular diameter Δs results in $D_{c}\propto \lambda \;L\;/\;\left(\pi \;\Delta s\right)$ while the coherence diameter of an incoherent source with Gaussian-distributed intensity and the standard deviation of a propagating coherent Gaussian source are both some multiple of $\lambda \;L\;/\;(\pi \;\sigma_{0})$. Therefore, in this article, it is assumed that $D_{c}$ is given by the beam width of the propagating Gaussian wave, which is approximated as $2\;\sqrt{2}\;\sigma_{D}$ for the far field, as shown in Figure 8b, where the intensity drops to $1/e^{2}$ at $D_{c}\;/\;2$. The accuracy of the estimation is further improved to include near field due to the diverse parameter ranges of the LGI and time domain interference setups as follows:(58)$$D_{c}\left(t,\sigma_{0}\right)\approx 2\;\sqrt{2}\;\sigma_{D}=\frac{2}{\sqrt{-A_{0}}}$$
where $\sigma_{D}\equiv 1\;/\;\sqrt{-2\;A_{0}}$ for $A_{0}<0$ since the free-space propagating Gaussian wave function on the detector plane $\Psi_{D}\left(x\right)$ is proportional to $\exp(A_{0}\;x^{2}\;+\;ı\;B_{0}\;x^{2})$, $A_{0}=-m^{2}\;\sigma_{0}^{2}\;/\;\left(2\;\hbar^{2}\;L^{2}\;/\;c^{2}\;+\;2\;m^{2}\;\sigma_{0}^{4}\right)$, and t≡L/c based on trivial application of FPI kernel in Equation (Equation 4). The slit positions on the first plane should be inside the coherence area for the proposed numerical simulations to be more compatible with the future experimental implementations.

In numerical simulations of $Sim_{1}$ for the violation of LGI with two planes of triple slits, the position of the furthest edge of a slit on PL-1 is found by $D_{s}+\;\Delta x\;\beta_{1}+\sqrt{2}\;\beta_{1}$ for $D_{s}>0$ by assuming that the Gaussian slit has a width of $2\;\sqrt{2}\;\beta_{1}$ similar to the Gaussian source formulation. Then, for the specific set of $\left(\sigma_{0},D_{s},\beta_{1},\Delta x\right)$ shown in Figure 6a,b, the simulation results are reliable if the slits are in the spatial coherence area formulated as follows:(59)$$D_{c}\left(t_{01},\sigma_{0}\right)\geq D_{1}\equiv 2\;\left(D_{s}+\;\left(\Delta x\;+\;\sqrt{2}\right)\;\beta_{1}\right)$$
where $D_{1}$ corresponds to the diameter of the area on PL-1, where the slits reside as shown in Figure 8c.

Similarly, the maximum difference between the positions of the slits on PL-1 and PL-2, i.e., the upper slit on PL-1 and the lower slit on PL-2, is calculated by using $D_{2}\equiv 2\;\left(D_{s}+\;\left(\Delta x\;+\;\sqrt{2}\right)\;\beta_{1}\right)+2\;\left(\Delta x\;+\;\sqrt{2}\right)\;\beta_{2}$. It is assumed that the projected light through the slit on the (j−1)th plane has spatial coherence larger than $D_{c}\left(t_{j-1,j},\beta_{j-1}\right)$ by assuming that the slit masking modifies the beam width to $\approx 2\;\sqrt{2}\;\beta_{j-1}$. More accurate analysis left as an open issue requires formulation of the coherence for the wave function $\psi_{j,n}\left(x_{j}\right)$ for each *n*th path in Equation (Equation 5) by also considering the parameter $A_{j}$ for calculating the beam width on the *j*th plane instead of utilizing $-m^{2}\;\beta_{j-1}^{2}\;/\;\left(2\;\hbar^{2}\;L^{2}\;/\;c^{2}\;+\;2\;m^{2}\;\beta_{j-1}^{4}\right)$ calculated by replacing $\sigma_{0}$ with $\beta_{j-1}$ in the expression of $A_{0}$. Then, the validity of the simulation requires the following inequality between PL-1 and PL-2:(60)$$D_{c}\left(t_{12},\beta_{1}\right)\geq D_{2}$$
Similar formulations for $Sim_{2}$ with three planes as shown in Figure 8d for the time domain interference result in the following requirements for the results in Figure 7 to be compatible with the sources practically not fully coherent: (61)$$\begin{array}{ccc} D_{c}\left(t_{01},\sigma_{0}\right) & \geq & D_{1}\equiv 2\;\left(\Delta x\;+\;\sqrt{2}\right)\;\beta_{1} \end{array}$$(62)$$\begin{array}{ccc} D_{c}\left(t_{12},\beta_{1}\right) & \geq & D_{2}\equiv 2\;\Delta x\;\beta_{1}\;+2\;X_{2,1}\;+\;2\;\sqrt{2}\;\beta_{2} \end{array}$$(63)$$\begin{array}{ccc} D_{c}\left(t_{23},\beta_{2}\right) & \geq & D_{3}\equiv 2\;\left(|X_{3,1}-X_{2,1}|\;+\;\sqrt{2}\beta_{3}\right) \end{array}$$

## Figures and Tables

**Figure 1 entropy-22-00246-f001:**
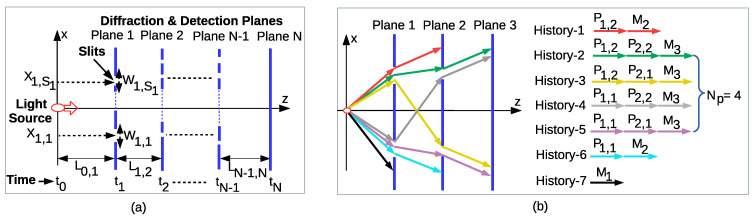
(**a**) System model of the free propagating light with velocity *c* in the *z*-direction and MPD through *N* planes, where *j*th plane includes $S_{j}$ slits at positions $X_{j,i}$ for $i\in [1,S_{j}]$ and interplane distance of $L_{j,j+1}$. (**b**) Example of three plane diffractions (N=3) with two slits for the first and second planes showing all the possible seven types of histories composed of diffractions or projections $P_{1,1}$, $P_{1,2}$, $P_{2,1}$, and $P_{2,2}$ through slits and measurements $M_{1}$, $M_{2}$, and $M_{3}$ on the planes. There are $N_{p}\equiv \prod_{j=1}^{N-1}S_{j}=2\times 2=4$ paths detected on the third plane.

**Figure 2 entropy-22-00246-f002:**
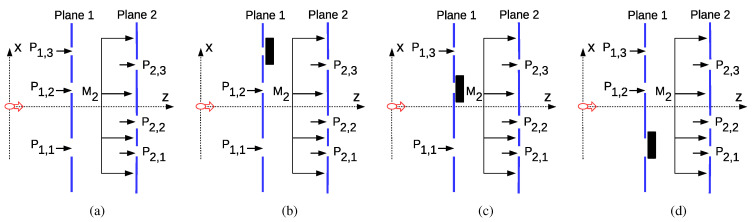
(**a**) The violation of Leggett–Garg Inequality (LGI) with the setup of two planes with triple slits where the event set at time $t_{1}$ is $\left[P_{1,1}\right]$, $\left[P_{1,2}\right]$, and $\left[P_{1,3}\right]$ and, at time $t_{2}$, are $\left[P_{2,1}\right]$, $\left[P_{2,1}\right]$, $\left[P_{2,3}\right]$, and $\left[M_{2}\right]$ and ambiguous measurement setups by closing (**b**) the third, (**c**) the second, and (**d**) the first slits on the first plane.

**Figure 3 entropy-22-00246-f003:**
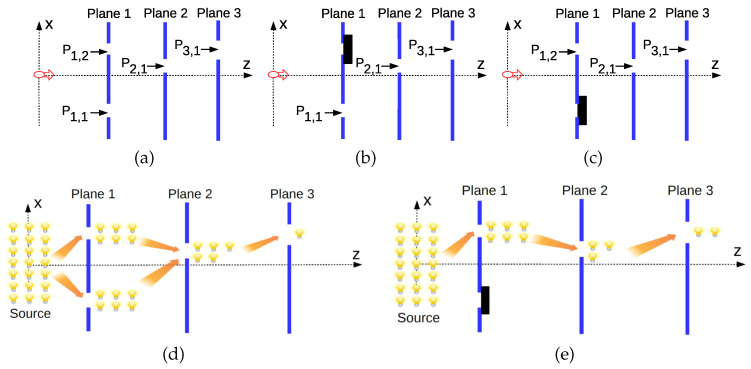
Setup for constructive and destructive interferences in time for the probabilities to diffract through each plane showing the history states (**a**) $|\Psi_{3}^{a})\equiv \left[P_{3,1}\right]⊙\left[P_{2,1}\right]⊙\left(\left[P_{1,1}\right]+\left[P_{1,2}\right]\right)⊙\left[\rho_{0}\right]$ as the superposition of $|\Psi_{3}^{b})$ and $|\Psi_{3}^{c})$, (**b**) $|\Psi_{3}^{b})\equiv \left[P_{3,1}\right]⊙\left[P_{2,1}\right]⊙\left[P_{1,1}\right]⊙\left[\rho_{0}\right]$, and (**c**) $|\Psi_{3}^{c})\equiv \left[P_{3,1}\right]⊙\left[P_{2,1}\right]⊙\left[P_{1,2}\right]⊙\left[\rho_{0}\right]$. The targeted scenario with *classically counterintuitive* nature where a specific example of interference pattern (represented as the number of lambs denoting the number of photons for a practical counting experiment) for the cases of (**d**) two slits on PL-1 both open and (**e**) only the second slit open. The operation of closing the first slit decreases the number of photons diffracted through PL-2 while counterintuitively increases the number of photons through PL-3 since we classically expect a decrease. This scenario shows the interference of histories at two different time instants for PL-2 and PL-3 with firstly constructive and then destructive effects, respectively.

**Figure 4 entropy-22-00246-f004:**
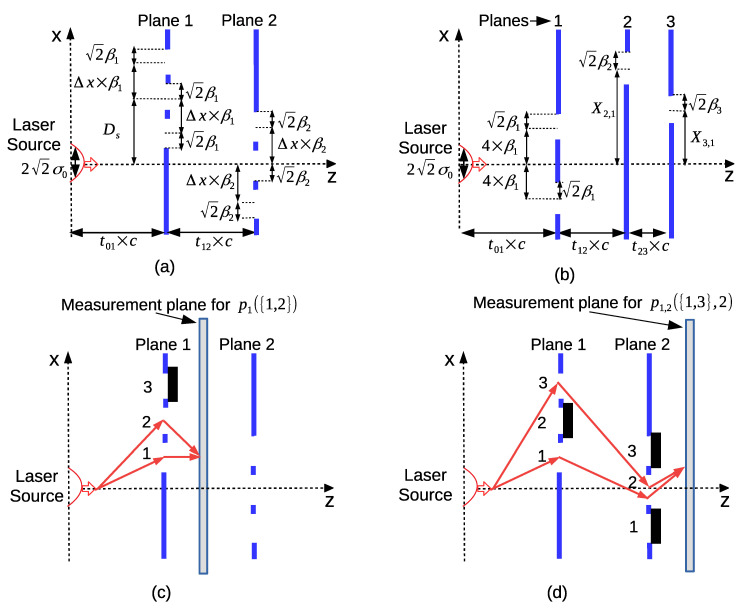
The layouts used in (**a**) $Sim_{1}$ and (**b**) $Sim_{2}$, where for $Sim_{2}$, the fixed values of the parameters are $\sigma_{0}=200$ (μm), $t_{01}=0.5$ (ns), $t_{12}=0.2$ (ns), $t_{23}=0.1$ (ns), $\beta_{1}=25$ (μm), $\beta_{2}=35$ (μm), and $\beta_{3}=45$ (μm) in addition to the fixed values of the slit positions on the first plane. The practical measurement setups to be utilized in future experiments are illustrated for the probabilities (**c**) $p_{1}\left(\left\{1,2\right\}\right)$ and (**d**) $p_{1,2}\left(\left\{1,3\right\},2\right)$. The measurement planes count the detected number of photons compared with the number of photons emitted by the source in unit time.

**Figure 5 entropy-22-00246-f005:**
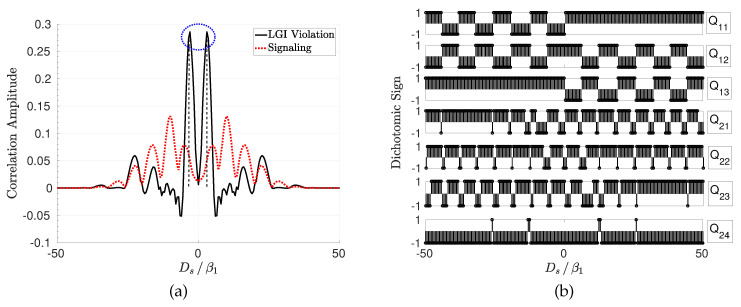
(**a**) LGI violation ($K_{A}\;-\;K_{V}$) and signaling ($K_{V}\;-1\;$) for varying $D_{s}$, where $t_{01}=0.2$ (ns), $t_{12}=0.1$ (ns), Δx=7, $\beta_{1}=15$ (μm), $\beta_{2}=30$ (μm), and $\sigma_{0}=130$ (μm) and (**b**) the corresponding dichotomic sign assignments for ambiguous measurements maximizing the violation for each $D_{s}$.

**Figure 6 entropy-22-00246-f006:**
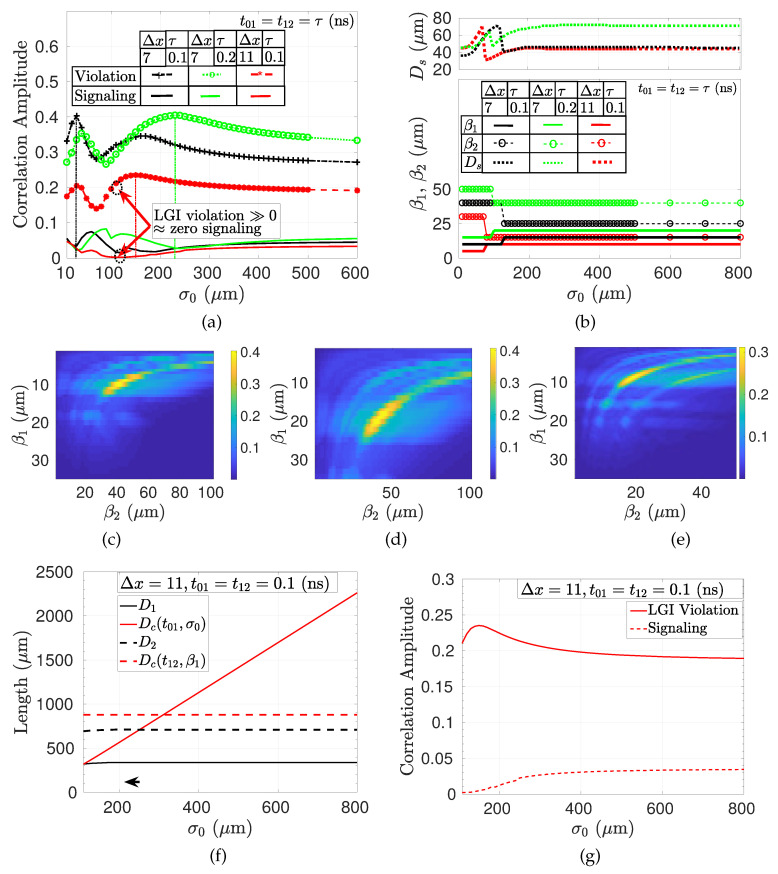
(**a**) Maximum LGI violation ($K_{A}-K_{V}$) and the corresponding amount of signaling ($K_{V}-1$) for varying $\sigma_{0}$, Δx, and $t_{01}=t_{12}$ and (**b**) the corresponding values of $\beta_{1}$, $\beta_{2}$, and $D_{s}$ maximizing the violation for each $\sigma_{0}$ assuming fully coherent sources. Maximum violation for varying $\left(\beta_{1},\beta_{2}\right)$ pairs for fully coherent sources where (**c**) Δx=7 and $t_{01}=t_{12}=0.1$ (ns) at the maximizing $\sigma_{0}=30$ (μm), (**d**) Δx=7 and $t_{01}=t_{12}=0.2$ (ns) at $\sigma_{0}=230$ (μm), and (**e**) Δx=11 and $t_{01}=t_{12}=0.1$ (ns) at $\sigma_{0}=150$ (μm). It is observed that there is a large set of slit pairs and beam width resulting in LGI violation reaching ≈0.4 for Δx=7 and ≈0.23 for Δx=11, respectively, while there are local peaks for $\left(\beta_{1},\beta_{2}\right)$ pairs for all cases. Increasing $t_{01},t_{12}$ values expands the $\left(\beta_{1},\beta_{2}\right)$ pairs for similar values of violations. (**f**) The comparison of the spatial coherence diameters $D_{c}$ with the diffraction setup diameters $D_{1}$ and $D_{2}$ for the first and second planes, respectively, where the targeted case is Δx=11 and $t_{01}=t_{12}=0.1$ (ns), i.e., analyzed as the red curve in Figure 6a, and (**g**) the corresponding LGI violation curve plotted again by emphasizing the coherence including the peak points.

**Figure 7 entropy-22-00246-f007:**
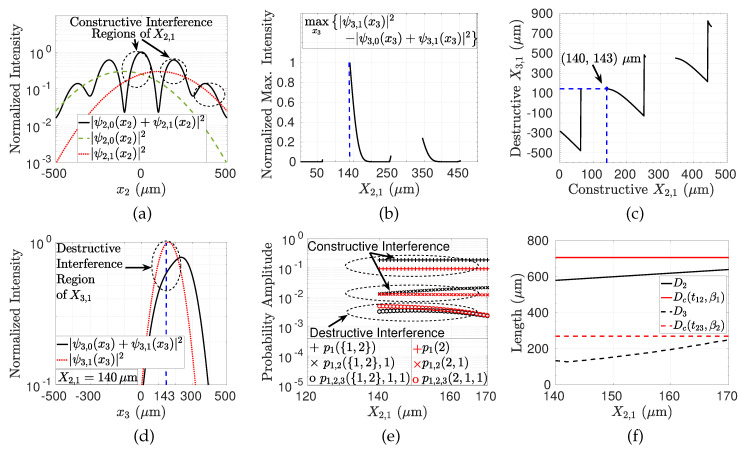
(**a**) $|\psi_{2,0}\left(x_{2}\right)\;+\;\psi_{2,1}\left(x_{2}\right){|}^{2}$ compared with $|\psi_{2,1}\left(x_{2}\right){|}^{2}$ and $|\psi_{2,0}\left(x_{2}\right){|}^{2}$ for diffraction through the layer PL-2, (**b**) $\max_{x_{3}}\left\{|\psi_{3,0}\left(x_{3}\right)\;+\;\psi_{3,1}\left(x_{3}\right){|}^{2}-{\left|\psi_{3,1}\left(x_{3}\right)\right|}^{2}\right\}$ for varying $X_{2,1}$ on PL-3 such that destructive interference is maximized for each $X_{2,1}$ with respect to $x_{3}$ while $X_{2,1}\approx 140\;\mu$m maximizes the destructive interference, (**c**) $X_{3,1}$ maximizing the destructive interference for varying $X_{2,1}$, (**d**) the comparison of $|\psi_{3,0}\left(x_{3}\right)\;+\;\psi_{3,1}\left(x_{3}\right){|}^{2}$ and $|\psi_{3,1}\left(x_{3}\right){|}^{2}$ on PL-3 for specific $X_{2,1}\approx 140\;\mu$m showing the destructive interference maximized with $X_{3,1}\approx 143\;\mu$m, and (**e**) the marked regions satisfy the counterintuitive scenario in (Equation 53)–(Equation 55) for varying $X_{2,1}$ with the corresponding $X_{3,1}$ pair in Figure 7c. Constructive and destructive interferences are observed for diffraction through PL-2 and PL-3, respectively, with different kinds of correlation of the paths at different times as a proof-of-concept numerical simulation of *quantum path interference (QPI) in time* between the two paths. (**f**) The comparison of setup diameters on the second and third planes, i.e., $D_{2}$ and $D_{3}$, respectively, with the spatial coherence diameters $D_{c}\left(t_{12},\beta_{1}\right)$ and $D_{c}\left(t_{23},\beta_{2}\right)$, respectively, in the targeted range of $X_{2,1}\in \left[140,170\right]$ (μm) in Figure 7e.

**Figure 8 entropy-22-00246-f008:**
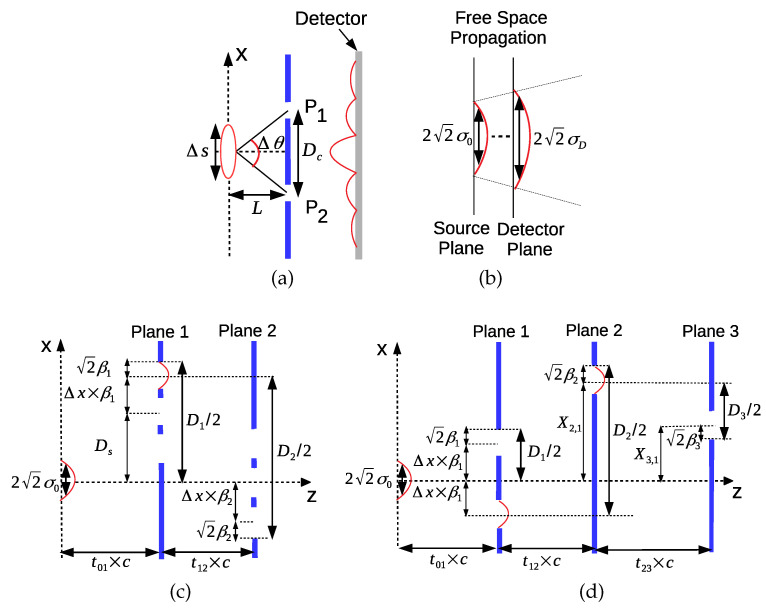
(**a**) The conventional modeling for the spatial coherence of light sources based on double-slit diffraction [43], where ΔθΔs≤λ is required for the fringes to be observed determining the spatial coherence diameter ($D_{c}$); (**b**) free-space propagation of Gaussian beam, where $D_{c}$ is approximated as the $1/e^{2}$ intensity beamwidth of $2\;\sqrt{2}\;\sigma_{0}$ with the standard deviation of $\sigma_{D}$. The descriptions of the calculation of the setup diameters on the planes to include the slits are denoted by $D_{j}$ for j∈[1,3] with respect to the location and the standard deviation of the source on the previous plane ($\sigma_{0}$ for the first plane and $\beta_{j-1}$ for the *j*th plane) for (**c**) LGI violation numerical analysis $Sim_{1}$ with two planes of triple slits on each plane and (**d**) interference in time scenario $Sim_{2}$ with three planes.

**Table 1 entropy-22-00246-t001:** Simulation and system parameters.

ID	Property	Value	ID	Property	Value
$Sim_{1}$	${\vec{X}}_{1}^{T}$	$D_{s}\;+\;\left[-\Delta x\;\;\;\;0\;\;\;\;\Delta x\right]\times \beta_{1}$	$Sim_{2}$	${\vec{X}}_{1}^{T}$	$\left[-4\;\;\;\;4\right]\times \beta_{1}$
	${\vec{X}}_{2}^{T}$	$\left[-\Delta x\;\;\;\;0\;\;\;\;\Delta x\right]\times \beta_{2}$		$X_{2,1}$ (μm)	$\left[0,\;500\right]$
	Δx; $D_{s}$	{7,11}; $\left[0,3000\right]$ (μm)		$X_{3,1}$ (μm)	$\left[-600,\;800\right]$
	$t_{01}=t_{12}$ (ns)	{0.1,0.2}		$t_{01},t_{12},t_{23}$ (ns)	0.5, 0.2, 0.1
	$\beta_{1}$, $\beta_{2}$ (μm)	$\left[1,\;50\right]$, $\left[1,\;100\right]$		$\beta_{1}$, $\beta_{2}$, $\beta_{3}$ (μm)	25, 35, 45
	$\sigma_{0}$ (μm)	$\left[10,\;800\right]$		$\sigma_{0}$ (μm)	200

**Table 2 entropy-22-00246-t002:** Parameters for modeling LGI and path integrals $(\psi_{2,i}\left(x_{2}\right)$ for i∈[0,2]: three paths).

	Formula		Formula		Formula
$k_{1}$	$\frac{-\hbar \;m\;t_{12}\left(a_{t,\sigma }\;+\;\hbar^{2}\;t_{01}\;t_{12}\right)}{2\;k_{11}}$	$k_{8}$	$-\frac{1}{4}\;\left(\frac{1}{\beta_{1}^{2}}\;+\;\frac{1}{\beta_{1}^{2}\;+\;d_{t,\sigma }}\right)$	$\xi_{1}$	$\frac{\beta_{1}^{2}\;m\;\vartheta_{t}}{\left(\beta_{1}^{2}\;m\;\left(m\;\sigma_{0}^{2}\;+\;ı\;\sqrt{b_{t}}\right)\;+\;ı\;\hbar \;t_{1,2}\;\vartheta_{t}\right)}$
$k_{2}$	$\hbar \;m^{3}\;\sigma_{0}^{2}t_{12}\left(\beta_{1}^{2}\;+\;d_{t,\sigma }\right)\;/\;k_{11}$	$k_{9}$	$\frac{1}{2}\;\left(\frac{1}{\beta_{1}^{2}}\;-\;\frac{1}{\beta_{1}^{2}\;+\;d_{t,\sigma }}\right)$	$A_{1}$	$\frac{-\beta_{1}^{2}\;m^{2}\left(\hbar^{2}\;t_{0,1}^{2}\;+\;m^{2}\;\sigma_{0}^{2}\;\Xi_{1}\right)}{\left(2\;\alpha_{t,\sigma ,\beta }\right)}$
$k_{3}$	$\left(-\beta_{1}^{2}m^{2}\left(a_{t,\sigma }\;+\;\hbar^{2}\;t_{01}\;t_{12}\right)\right)\;/\;k_{11}$	$k_{10}$	$-1\;/\;(\beta_{1}^{2}\;+\;d_{t,\sigma })$	$B_{1}$	$\frac{\left(\beta_{1}^{4}\;m^{3}\;\hbar \;t_{0,1}\;+\;m\;\hbar \;t_{1,2}\left(\hbar^{2}\;t_{0,1}^{2}\;+\;m^{2}\;\Xi_{1}^{2}\right)\right)}{\left(2\;\alpha_{t,\sigma ,\beta }\right)}$
$k_{4}$	$\beta_{1}^{2}\;m^{4}\;\sigma_{0}^{2}\left(\beta_{1}^{2}\;+\;d_{t,\sigma }\right)\;/\;k_{11}$	$k_{11}$	$\begin{array}{l} \beta_{1}^{4}\;m^{2}\;\left(m^{2}\;\sigma_{0}^{2}\;\Xi_{2}\;+\;b_{t}\;\right)\\ +\;\beta_{1}^{2}m^{2}\left(\beta_{2}^{2}\;a_{t,\sigma }\;+\;2\;c_{t,\sigma }\right)\\ +\;\hbar^{2}\;t_{12}^{2}\;a_{t,\sigma } \end{array}$	$H_{R,1}$	$\frac{-m^{2}\left(\beta_{1}^{2}\left(b_{t}\;+\;m^{2}\sigma_{0}^{4}\right)\;+\;c_{t,\sigma }\right)}{\left(2\;\alpha_{t,\sigma ,\beta }\right)}$
$k_{5}$	$\frac{-m^{2}\left(\beta_{1}^{2}\left(m^{2}\;\sigma_{0}^{2}\;\Xi_{2}\;+\;b_{t}\right)\;+\;c_{t,\sigma }\right)}{k_{11}}$	$\begin{array}{l} a_{t,\sigma }\\ b_{t}\\ c_{t,\sigma } \end{array}$	$\begin{array}{l} \hbar^{2}\;t_{01}^{2}\;+\;m^{2}\sigma_{0}^{4}\\ \hbar^{2}\;{\left(t_{01}\;+\;t_{12}\right)}^{2}\\ \hbar^{2}\;\sigma_{0}^{2}\;t_{12}^{2} \end{array}$	$H_{I,1}$	$\frac{m\;\hbar \;t_{1,2}\left(a_{t,\sigma }\;+\;\hbar^{2}\;t_{0,1}\;t_{1,2}\right)}{\left(2\;\alpha_{t,\sigma ,\beta }\right)}$
$k_{6}$	$\begin{array}{l} -m^{2}\left(2\;\beta_{1}^{2}\;\left(m^{2}\;\sigma_{0}^{2}\;\Xi_{2}\;+b_{t}\right)\right)\;/\;\left(4\;k_{11}\right)\\ +\;\;m^{2}\left(-\beta_{2}^{2}\;a_{t,\sigma }\;-\;2\;c_{t,\sigma }\right)\;/\;\left(4\;k_{11}\right) \end{array}$	$\alpha_{t,\sigma ,\beta }$	$\begin{array}{l} \beta_{1}^{4}m^{2}\left(b_{t}\;+\;m^{2}\sigma_{0}^{4}\right)\;\\ +\;2\;\beta_{1}^{2}\;m^{2}\;c_{t,\sigma }\\ +\;\hbar^{2}\;t_{1,2}^{2}\;a_{t,\sigma } \end{array}$	$c_{1}$	$\beta_{1}^{2}\;m^{2}\left(a_{t,\sigma }\;+\;\hbar^{2}\;t_{0,1}\;t_{1,2}\right)\;/\;\alpha_{t,\sigma ,\beta }$
$k_{7}$	$\beta_{2}^{2}\;m^{2}\;a_{t,\sigma }\;/\;\left(2\;k_{11}\right)$	$\chi_{0}$	$\pi^{-1/4}\sqrt{\frac{m\;\sigma_{0}}{m\;\sigma_{0}^{2}\;+\;ı\;\hbar \;t_{0,1}}}$	$d_{1}$	$\frac{-m\;\hbar \;t_{1,2}\left(\hbar^{2}t_{0,1}^{2}\;+\;m^{2}\;\sigma_{0}^{2}\;\Xi_{1}\right)}{\alpha_{t,\sigma ,\beta }}$

**Table 3 entropy-22-00246-t003:** Parameters for modeling the path integrals of QPI $(\psi_{3,i}\left(x_{3}\right)$ for i∈[0,1]: two paths).

	Formula		Formula	j∈[1,2]	Formula
$H_{2}$	$\left(\begin{array}{cc} \nu_{2,2}{\left(\zeta_{1,c}\;+ı\;\zeta_{1,d}\right)}^{2}\;+\;\nu_{1,1} & 0\\ \nu_{3,2}\left(\zeta_{1,c}\;+ı\;\zeta_{1,d}\right) & \nu_{1,2} \end{array}\right)$	$\nu_{2,2}$	$-\frac{\beta_{2}^{2}\;\hbar \;t_{2,3}}{2\;ı\;\varsigma_{2}}$	$\nu_{1,j}$	$-\frac{2\;\hbar \;t_{j,j+1}\left(A_{j-1}\;+\;ı\;B_{j-1}\right)\;+\;ı\;m}{2\;ı\;\varsigma_{j}}$
		$\nu_{3,2}$	$-\frac{\hbar \;t_{2,3}}{ı\;\varsigma_{2}}$	$\zeta_{j}$	$\begin{array}{l} 4\;B_{j-1}\;\beta_{j}^{4}\;\hbar \;m\;t_{j,j+1}\;+\;\beta_{j}^{4}\;m^{2}\;\\ +\;\hbar^{2}\;t_{j,j+1}^{2}\;\varrho_{j} \end{array}$
$\begin{array}{ll} {\vec{c}}_{2} & \left(\begin{array}{c} \;\nu_{4,2}\;\zeta_{1,c}\;+\;\nu_{5,2}\;\zeta_{1,d}\\ \zeta_{2,c} \end{array}\right)\\ {\vec{d}}_{2} & \left(\begin{array}{c} \nu_{4,2}\;\zeta_{1,d}\;-\;\nu_{5,2}\;\zeta_{1,c}\\ \zeta_{2,d} \end{array}\right) \end{array}$		$\nu_{4,2}$	$\beta_{2}^{2}\;\zeta_{2,c}$	$\zeta_{j,c}$	$\left(2\;B_{j-1}\;\hbar \;m\;t_{j,j+1}\;\beta_{j}^{2}+\beta_{j}^{2}\;m^{2}\right)\;/\;\zeta_{j}$
		$\nu_{5,2}$	$-\frac{2\;\hbar \;t_{2,3}\;A_{2}}{m}$	$\zeta_{j,d}$	$\hbar \;m\;t_{j,j+1}\;\left(2\;A_{j-1}\;\beta_{j}^{2}\;-\;1\right)\;/\;\zeta_{j}$
$A_{0}$	$-m^{2}\sigma_{0}^{2}\;/\;\left(2\;\hbar^{2}t_{0,1}^{2}\;+\;2\;m^{2}\sigma_{0}^{4}\right)$	$A_{2}$	$\frac{\beta_{2}^{2}\;m^{2}\;\left(2\;A_{1}\;\beta_{2}^{2}\;-\;1\right)}{2\;\zeta_{2}}$	$\begin{array}{l} \varrho_{j}\\ \xi_{j} \end{array}$	$\begin{array}{l} 4\;\beta_{j}^{4}\left(A_{j-1}^{2}+B_{j-1}^{2}\right)\;-\;4\;A_{j-1}\beta_{j}^{2}+1\\ \beta_{j}^{2}\;m\;/\;\varsigma_{j} \end{array}$
$B_{0}$	$\hbar \;m\;t_{0,1}\;/\;\left(2\;\hbar^{2}t_{0,1}^{2}\;+\;2m^{2}\sigma_{0}^{4}\right)$	$B_{2}$	$\frac{2\;B_{1}\;\beta_{2}^{4}\;m^{2}+\hbar \;m\;t_{2,3}\;\varrho_{2}}{2\;\zeta_{2}}$	$\varsigma_{j}$	$\begin{array}{l} \hbar \;t_{j,j+1}\;\left(2\;\beta_{j}^{2}\;\left(B_{j-1}-\;ı\;A_{j-1}\right)\;+\;ı\right)\\ \;+\beta_{j}^{2}\;m \end{array}$

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
