# Peer review of "Theory of Quantum Path Entanglement and Interference with Multiplane Diffraction of Classical Light Sources"

_entropy, 2020, doi:10.3390/e22020246_

Round 1

Reviewer 1 Report

The authors apply the consistent histories theory for entangled states. In particular, they show how to exploit the tensor product structure that arises from the consistent histories with the consistency condition for quantum computing resource by performing an exhaustive study of the multi-plane diffraction of photons. The manuscript is exhaustive in the sense that the theory and the numerical implementation are explained with details. The original idea that consists of the generalization of the quantum histories theory for entangled states is interesting for physics and for philosophers of science. The manuscript is well written and with a careful read can be understood. The conclusions are correct and the references used in the manuscript are of high impact. For all this I consider this manuscript suitable for publication. The unique observation is that in other works about quantum histories, there is a "contextual condition" that is used instead of a "consistent condition" (see 10.1016/j.aop.2014.03.001 and 10.1007/s11128-018-2050-3) that perhaps require an explanation from the author but I do not consider this against publication or minor revision.

Author Response

The response letter is attached.

Reviewer 2 Report

Please incorporate the layout of the experiments with all parameters.

Author Response

The response letter is attached for both the reviewer comments.
